
# Residential building stock modeling for mainland China targeted for seismic risk assessment

Danhua Xin[1,2,*], James Edward Daniell[2,3,*], Hing-Ho Tsang[4], Friedemann Wenzel[2]

[1]Department of Earth and Space Sciences, Southern University of Science and Technology, 1088 Xueyuan Avenue,
Shenzhen 518055, Guangdong Province, China

[2]Center for Disaster Management and Risk Reduction Technology (CEDIM) and Geophysical Institute, Karlsruhe
Institute of Technology, Hertzstrasse 16, 76187, Karlsruhe, Germany

[3]General Sir John Monash Scholar, The General Sir John Monash Foundation, Level 5, 30 Collins Street,
Melbourne, Victoria, 3000, Australia

[4]Centre for Sustainable Infrastructure, Swinburne University of Technology, Melbourne, VIC 3122, Australia

*Correspondence to James Edward Daniell (j.e.daniell@gmail.com), Danhua Xin (xindh@sustech.edu.cn)

## Abstract

Previous seismic damage reports have shown that the damage and collapse of buildings is the leading cause of fatality and property loss. To enhance the estimation accuracy of economic loss and fatality in seismic risk assessment, a high-resolution building exposure model is important. Previous studies in developing global and regional building exposure models usually use coarse administrative level (e.g., county, or sub-country level) census data as model inputs, which cannot fully reflect the spatial heterogeneity of buildings in large countries like China. To develop a high-resolution residential building stock model for mainland China, this paper uses finer urbanity level population and building-related statistics extracted from the records in Tabulation of the 2010 Population Census of the People's Republic of China (hereafter abbreviated as the "2010-census"). In the 2010-census records, for each province, the building-related statistics are categorized into three urbanity levels (urban, township, and rural). Statistics of each urbanity level are from areas with a similar development background but belong to different administrative prefectures and counties. Due to privacy protection-related issues, these urbanity level statistics are not geo-coded. Therefore, before disaggregating these statistics into high-resolution grid level, we need to determine the urbanity attributes of grids within each province. For this purpose, the geo-coded population density profile (with 1km×1km resolution) developed in the 2015 Global Human Settlement Layer (GSHL) project is selected to divide the 31 provinces of mainland China into 1km×1km grids. Then for each province, the grids are assigned with urban/township/rural attributes according to the population density in the 2015 GHSL profile. Next for each urbanity of each province, the urbanity level building-related statistics extracted from the 2010-census records can be disaggregated into the 2015 GHSL geo-coded grids, and the 2015 GHSL population in each grid is used as the disaggregation weight. Based on the four structure types (steel/reinforced-concrete, mixed, brick/wood, other) and five storey classes (1, 2-3, 4-6, 7-9, ≥10) of residential buildings classified in the 2010-census records, we reclassify the residential buildings into 17 building subtypes attached with both structure type and storey class and estimate their unit construction prices. Finally, we develop a geo-coded 1km×1km resolution residential building exposure model for 31 provinces of mainland China. In each 1km×1km





grid, the floor areas of the 17 residential building subtypes and their replacement values are estimated. To evaluate the model performance, comparisons with the wealth capital stock values estimated in previous studies at the administrative prefecture-level and with the residential floor area statistics in the 2010-census at the administrative county/prefecture-level are conducted. The practicability of the modeled results in seismic risk assessment is also

checked by estimating the seismic loss of residential buildings in Sichuan Province combined with the intensity map of the 2008 Wenchuan Ms8.0 earthquake and an empirical loss function developed from historical seismic damage information in China. Our estimated seismic loss range is close to that derived from field investigation reports. Limitations of this paper and future improvement directions are discussed. More importantly, the whole modeling process of this paper is fully reproducible, and all the modeled results are publicly accessible. Given that

the building stock in China is changing rapidly, the results can be conveniently updated when new datasets are available.

Key Words: residential building stock modeling, 2010-census records, dasymmetric disaggregation

## 1.    Introduction

The frequent occurrence of earthquakes and other natural hazards (typhoon, flood, tsunami, etc.) can lead to

tremendous and often crippling economic losses. According to the estimation in Daniell et al. (2017), from 1900-2016, 2.3 million earthquake fatalities from 2233 fatal events occurred worldwide. Economic losses (direct and indirect) associated with the occurrence of over 9,900 damaging earthquakes reached USD 3.41 trillion (in 2016 prices). For cases in China, the combination of high seismic activity, population density, and building vulnerability cause even higher seismic risk: Earthquakes that occurred in China during the 110 years from 1900 to 2010

accounted for about 2.5% of radiated energy globally, but the earthquake fatality ratio is around 1/3 of the world (Wu et al., 2013). Among the losses caused by natural disasters, buildings are considered as the most important asset category, since the main sources of loss and fatality that occurs during earthquakes are related to building damage and collapse (e.g., Neumayer and Barthel, 2011; Yuan, 2008). Information on the exposed value of buildings is key to seismic loss estimation, whose accuracy will further affect the effectiveness in earthquake

response and rescue (Xu et al., 2016a). Therefore, in any seismic risk mitigation effort, the estimation of the building stock and the values at risk should be given top priority. This is even more urgent for seismic active and disaster vulnerable countries like China (Allen et al., 2009), where rapid urbanization has led to a massive increase in both the asset value and population that are exposed to a potential seismic hazard (Hu et al., 2010; Yang and Kohler, 2008).

Modeling seismic loss to buildings requires quantifying their exposure in terms of floor area and monetary value (Paprotny et al., 2020). A series of micro-, meso- and macro-scale approaches have been developed for this purpose. The scale of the method depends not only on the size of the study area but also on the goal of the investigation, the availability of necessary data, time, money, and human resources (Messner and Meyer 2006). For example, micro-scale analyses calculate the asset value based on individual buildings, which requires detailed information

on building characteristics (e.g., occupancy, age, structure type, building height, or the number of floors). However, since great efforts and considerable expenses are required to collect such information for each building, micro-scale methods are rarely applicable on a regional or (inter)national level (e.g., Figueiredo and Martina, 2016; Erdik, 2017). When further limited by the privacy protection issue, information on asset values of individual buildings is


more difficult to obtain (Wünsch et al., 2009). In contrast, meso- and macro-scale methods that use aggregated

exposure data on building characteristics procured from official statistics and organized in administrative units (e.g., country, province, prefecture, county/district, etc.) are more commonly used in modeling building values exposed to future earthquakes.

Since building-related statistics are usually aggregated at a coarse administrative level, while seismic hazards are usually modeled with high spatial resolution, there is a spatial mismatch between exposure data and hazard

mapping (e.g., Chen et al., 2004; Thieken et al., 2006). This mismatch may delay and mislead the recuse decision-making after large earthquakes. For example, after the occurrence of the Ms8.0 Wenchuan earthquake, one of the most severely affected areas, Qingchuan County, did not get an appropriate rescue response, while most of the recuse resources were sent to the less damaged Dujiangyan City. The major reason for this problem was: The exposure data (population, buildings) used to assess seismic loss were based on administrative units (Xu et al.,

2016). Therefore, to enhance seismic risk assessment accuracy, the aggregated building statistics data need to be spatialized into high-resolution grids levels. Several interpolation and decomposition methods (e.g., areal weighting, pycnophylactic interpolation, dasymetric mapping) have been developed for this purpose. Compared with the areal weighting method, in which the aggregated building data are evenly distributed (e.g., Goodchild et al. 1993), pycnophylactic interpolation method uses a smoothing function of distance to determine the

disaggregation weight (e.g., Tobler, 1979) and tends to be more reasonable, since the distribution of buildings within an administrative unit is heterogeneous. Based on the pycnophylactic interpolation method, the dasymetric mapping method (Bhaduri et al., 2007) further utilizes finer resolution ancillary spatial data to augment the interpolation process and is now widely used.

When using the dasymetric mapping method to spatialize the administrative level building exposure data, the

selection of appropriate ancillary information is thought to be the most difficult part (Wu et al., 2018), since such information should not only be geo-coded and readily available but also have a high correlation with the building exposure data to be disaggregated. A range of remote sensing data (e.g., nightlight data, road density, land use/land type, population spatial distribution datasets, etc.) has been employed as ancillary information in the literature. A detailed summary of these ancillary data will be given in the Data Sources and Methodology section.

Based on the aggregated building-related statistics and using the dasymetric mapping method, this paper develops a high-resolution residential building model (in terms of building floor area and replacement value) for seismic risk assessment in mainland China. This issue has been explored in many previous studies and a series of global and regional building exposure models have been developed. One famous such global model is the PAGER (Prompt Assessment of Global Earthquakes for Response) building inventory database, which is the first open,

publicly available, transparently developed global model (Jaiswal et al., 2010). However, the PAGER inventory was developed to rapidly estimate human occupancies in different structure types for earthquake fatality assessment. It lacks information in actual building counts and does not use available information from a commercial database or remote sensing data, thus cannot be used for building asset evaluation immediately (Dell'Acqua et al., 2013). To overcome this difficulty, at least partially, the GED4GEM (the Global Exposure

Database for the Global Earthquake Model) project develops a complementary approach that can provide a spatial inventory of exposed assets for catastrophe modeling and loss estimation worldwide (Gamba, 2014). The input datasets ingested into the GED4GEM are at multiple spatial scales, from coarse country-level statistics to finer





compilations of each building in some sample regions. There are also other global models, such as the series of building stock models released by the Global Assessment Report (De Bono and Chatenous, 2015; De Bono and

Mora, 2014; De Bono et al., 2013) of the United Nations International Strategy for Disaster Reduction (UNISDR), and the global exposure dataset created by Gunasekera et al. (2015). When focusing on the modelling of building stock in China, a common limitation shared by these global models is that the building-related statistics they disaggregate are only of country/sub-country level, although finer level statistics are already available. Thus, a general assumption in the disaggregation process of these global models is that building stock value per capita

within the country/sub-country is uniform. A similar assumption is also made in studies that develop building exposure models specifically for China (e.g., Yang and Kohler, 2008; Hu et al., 2010). For computational convenience, such an assumption is acceptable. However, for improving the seismic risk assessment accuracy in each specific country, more detailed aggregated data at a finer level, if available, should be fully employed in the development of their building exposure model.

By considering the depreciation of all physical fixed assets (including residential and non-residential buildings, infrastructures, tools, machinery, and equipment), Wu et al. (2014) estimated the wealth capital stock (WKS) value for 344 prefectures in mainland China using the perpetual inventory method (PIM). Later, Wu et al. (2018) decomposed the prefecture-level WKS value into building assets, infrastructure assets, and other assets with fixed percentage shares of 44%, 19%, and 37% for all 344 prefectures. And these three asset components were further

disaggregated into 800m×800m high-resolution grids by using LandScan population, road density, and nighttime light as ancillary information, respectively. The basic idea of combining the use of different ancillary information to disaggregate the WKS value in Wu et al. (2018) is good. However, the over-simplification in fixing the percentage shares of the building, infrastructure, and other assets in all prefectures limits the applicability of their results in actual seismic risk assessment.

Based on the county-level building-related statistics extracted from the 2010-census records, Xu et al. (2016b) developed the nation-wide dasymetric foundation data (including population and buildings) for quick earthquake disaster loss assessment and emergency response in China by using the multi-variate regression method (Xu et al., 2016a). The multivariate regression method used in Xu et al. (2016a) was explained in more detail by Chen et al. (2012) and Han et al. (2013), in which they developed the population and building exposure models for areas in

Yunnan Province. Fu et al. (2014a) also used the multi-variate regression method to produce the 1km×1km resolution population grids in the years 2005 and 2010 for mainland China. Important assumptions in this multivariate regression method are: (1) The spatial distribution of population is limited within the six land use types (namely cultivated land, forest land, grass land, rural residential land, urban residential land, industrial and transportation land) recognized from the Landsat TM images; (2) For counties with similar geographical and

demographic characteristics (e.g., population number, structure and economy development level), the population density within each land use type is the same. Recently, Lin et al. (2020) conducted a township/street level comparison of population models generated by Fu et al. (2014a) and other institutes for Guangdong Province, China with the surveyed population in 2010-census records. Their comparison shows that the township/street level population generated by using the multi-variate method in Fu et al. (2014a) tends to overpredict the population

density in a sparsely populated area and underpredict the population density in densely populated area, especially the downtown area of metropolitan cities like Shenzhen and Guangzhou. The reason for these discrepancies is obvious: Since the population density developed for each land use type by using the multi-variate method is the





average population density. Although the building exposure model developed by Xu et al. (2016b) has not yet been tested, we conclude that the model of Xu et al. (2016b) also suffers from the over/under prediction problem in Fu et al. (2014a).

To overcome the limitations in building exposure models developed for mainland China in previous studies, this paper aims to present an improved method for generating a high-resolution residential building stock model (in terms of building floor area and replacement value) for mainland China. The main improvements in this paper are: (1) Compared with global building exposure models, we will use finer urbanity level (urban, township and rural) building related statistics extracted from the 2010-census records as model inputs; (2) Compared with Wu et al. (2018), in which the building assets are decomposed from the composite WKS value with fixed percentage share for all prefectures, we will use statistics that are directly related to residential buildings for each urbanity level of each province; (3) Compared with Xu et al. (2016b), in which only land use data are employed in the multi-variate method to derive the average building floor area density within each grid, we will use the ancillary population density profile generated from the 2015 Global Human Settlement Layer (GHSL), which is considered to be the best available assessment of spatial extents of human settlements with unprecedented spatial-temporal coverage and detail (e.g., Freire et al., 2016).

The organization of the paper is as follows. Sect. 2 (Data Sources and Methodology) will firstly describe the building-related statistics to be used as model inputs that extracted from the 2010-census records (Sect. 2.1), the review and selection of ancillary data to disaggregate these statistics into grid level (Sect. 2.2), and the derivation of residential building floor area and replacement value in each grid based on these statistics and the ancillary data (Sect. 2.3 and 2.4). Then the major results will be presented (Sect. 3.1) and comparisons with other independent data sources will be conducted (Sect. 3.2). Limitations in this paper and further improvement directions will also be discussed in Sect. 4.  Conclusions will be drawn in Sect. 5.

## 2.    Data Sources and Methodology

In dasymetric mapping, the use of finer scale census data as input and the choice of appropriate ancillary remote sensing data to disaggregate the census data into a higher grid level are the two controlling factors for the quality of the building stock model. For China, after the 2010 Sixth Population Census (namely the 2010-census), detailed statistical data related to residential building characteristics (e.g., building occupancy, structure type, height classes, etc.) are available for each province at the urbanity level (urban/township/rural). These urbanity level building-related statistics are good data sources to develop the building exposure model for China. To disaggregate these statistics into grid level, the correlation between the ancillary remote sensing data and the building-related statistics needs to be established. Then, the building floor area and replacement value at the grid level can be estimated. Therefore, in this section we will introduce the residential building-related statistics as extracted from the 2010-census records, the review/selection of ancillary remote sensing data to disaggregate these statistics into grid level, and the method to derive the grid level residential building floor area and replacement value based on these statistics and the ancillary remote sensing data.


### 2.1 The building-related statistics in the 2010-census records

The statistics to be used in this paper for building stock modeling are extracted from the Tabulation of the 2010 Population Census of the People's Republic of China (namely the 2010-census) particularly for residential buildings. Like in most countries of the world, the nation-wide population and housing census in China are carried out at the 10-year interval. The census for the year 2020 is just initiated and normally it takes around two years to publish the final surveyed data. Therefore, the current latest census data are for the year 2010. In the 2010-census, there are two types of tables: Long Table and Short Table. Long Table includes summaries based on the surveys of 10% of the total population in mainland China, while the Short Table summaries are based on the surveys of the whole population. Statistics on building characteristics (e.g., building occupancy type, height classes, structure type, etc.) are extracted from the Long Table of the 2010-census. Supplementary demographic statistics (e.g., the total population in each urbanity, the average number of people per family, and average floor area per person) are extracted from the Short Table of the 2010-census. A detailed introduction of corresponding sources of these data is given in Table 1.

For each of the 31 provincial administrative units in mainland China (including five autonomous regions: Xinjiang, Tibet, Ningxia, Inner Mongolia, Guangxi; and four municipalities: Beijing, Shanghai, Tianjin, Chongqing; hereafter all referred to as provinces), statistics on building characteristics in the Long Table of the 2010-census are aggregated into three urbanity levels (urban/township/rural). The urbanity attribute is determined according to the administrative unit of the surveyed population. As listed in Table 2, these statistics will be used as model inputs to develop the grid level residential building model in terms of floor area and replacement value. Compared with country/sub-country level census data used in previous global or regional models, the further categorization of building-related statistics into urbanity level in the 2010-census helps differentiate the spatial heterogeneity of buildings within each province, since the building-related statistics of the same urbanity level are from areas with similar development background but different administrative units. The spatial administrative boundaries used in this paper are from the National Geomatics Centre of China (see Data/Code Availability section for access).

### 2.2 Review/Selection of ancillary remote sensing data for dasymetric building stock modeling

Before disaggregating the urbanity level building-related statistics into 1km×1km grid level, appropriate ancillary information needs to be carefully selected and evaluated. The use of remote sensing data as ancillary information to determine the disaggregation weight is common in dasymetric modeling and has been frequently adopted in previous studies (e.g., Aubrecht et al. 2013; Gunasekera et al., 2015; Silva et al., 2015). The most commonly used remote sensing data include land use/land cover data (LULC, e.g., Eicher and Brewer, 2001; Wünsch et al., 2009; Seifert et al., 2010; Thieken et al., 2006), nighttime light data (e.g., Doll et al 2006; Ghosh et al, 2010; Chen and Nordhaus 2011; Ma et al., 2012) and road density data (e.g., Gunasekera et al., 2015; Wu et al., 2018). According to Wu et al. (2018), the LULC, nighttime light, road density data can be categorized as primary remote sensing data.

Each primary remote sensing data has its pros and cons when used for dasymetric disaggregation. For example, studies using LULC data (e.g., Globcover, GLC2000, MODIS, GlobeLand30) assume the population within each land-use type is uniformly distributed, which is a better assumption compared with believing in an evenly distributed population within an administrative unit. But this assumption is not consistent with the real situation.


(Thieken et al., 2006), specifically in suburban and rural areas, where the dispersion of population is greater than in urban areas (Bhaduri et al., 2007). Therefore, LULC data is inadequate to fully reflect the spatial heterogeneity within each land use or land cover class. In contrast, nighttime light data, acquired by the U.S. Air Force Defense Meteorological Satellite Program (DMSP) Operational Linescan System (OLS) (Elvidge et al., 2007) and provided

by the National Oceanic and Atmospheric Administration (NOAA) every year, are considered the most suitable ancillary information for indicating both the distribution and the density of human settlements and economic activities (Wu et al., 2018). Nighttime light data have been widely used to produce grid-based global population and GDP data sets (e.g., Ghosh et al, 2010; Chen and Nordhaus 2011; Ma et al., 2012). However, the drawbacks of nighttime light intensity data are also obvious. Limited by the operating conditions of DMSP satellites, the range

of nighttime light density is within a narrow interval of 0-63, thus leading to the pixel oversaturation in urban centers (Elvidge et al., 2007). For areas other than city centers (e.g., mountainous rural area), the coverage of nighttime light data is incomplete as it cannot correctly reflect the distribution of nonluminous objects (e.g., road transportation facilities, electricity infrastructure). Compared with the LULC and nighttime light data, road distribution data are more frequently used for assessing infrastructure assets, since power lines, energy pipelines,

water supply, and sewage pipelines are generally buried along the roads (Wu et al., 2018). Currently, road density data can be converted from road networks like OpenStreetMap, which is an openly available but crowdsourced online database (Zhang et al., 2015). As these data are not systematically compiled, there is still room for improvements (Wu et al., 2018).

Given the limitation of each primary remote sensing data, a series of secondary ancillary datasets are developed

based on the combined use of these primary datasets. For example, the famous LandScan population density profile was produced by apportioning the best available census counts into cells based on probability coefficients, which were derived from road proximity, slope, land cover, and night-time lights (Dobson et al., 2000). Based on these primary and secondary ancillary datasets, a series of studies have been conducted to disaggregate administrative level building census data into geo-coded grids. For example, Silva et al. (2015) disaggregated the building stock

at parish level for mainland Portugal based on the population density profile at 30×30 arc-sec resolution cells from LandScan. Gunasekara et al. (2015) developed an adaptive global exposure model (including three independent geo-referenced databases, namely building inventory stock, non-building infrastructure, and sector-based GDP), in which build-up area and LandScan population density are used to disaggregate country-level exposed asset value. Wu et al. (2018) established a high-resolution asset value map for mainland China by spatializing the

prefecture-level depreciated capital stock value into girds using the combination of three ancillary datasets— nighttime light, LandScan population, and road density, to name just a few.

In this paper, we follow the assumption of Thieken et al. (2006) that the distribution of residential asset values can be directly reflected by population distribution. Now the remaining question is to select appropriate ancillary population spatial distribution data to disaggregate building-related statistics in the 2010-census records. The

candidate population datasets include Gridded Population of the World (GPW, Balk and Yetman, 2004), Global Rural-Urban Mapping Project (GRUMP) population (see section Data/Code Availability), LandScan (Bhaduri et al., 2007), WorldPop (Linard et al., 2012) or AsiaPop (Gaughan et al., 2013), PopGrid China (Fu et al., 2014b), Global Human Settlement Layer (GHSL) population grids (Freire et al., 2016; Pesaresi et al., 2013) etc. GPW is a product of simple areal weighting interpolation and GRUMP is derived through simple dasymetric modeling, while

LandScan is structurally a multidimensional dasymetric model (Bhaduri et al., 2007). According to Gunasekera et



al. (2015), the LandScan gridded population dataset was identified as the best-suited dataset for exposure disaggregation, while other gridded population datasets such as GPW and GRUMP were too coarse in resolution and accuracy. According to Wu et al. (2018), LandScan, AsiaPop, and PopGrid China are the most promising population density datasets for asset value disaggregation in China since they all contain high-resolution attributes.

However, some population data of China are missing from the current AsiaPop. And compared with LandScan, the spatial coverage of PopGrid China is limited. Thus, the LandScan dataset was used for the final disaggregation of building assets in Gunasekera et al. (2015) and Wu et al. (2018). However, due to its commercial nature, the details to create the LandScan population datasets are less transparent, although being considered as one of the best global population density data sets (Sabesan et al., 2007). In contrast, the population datasets developed by

the GHSL project of the European Commission based on the global human settlement areas extracted from multi-scale textures and morphological features are transparent and freely available. The built-up area in GHSL was built by combining the MODIS 500 Urban Land Cover (MODIS500) and the LandScan 2010 population layer and are among the best-known binary products based on remote sensing (Ji et al., 2020). Preliminary tests confirm that the quality of the information on built-up areas delivered by GHSL is better than other available global information

layers extracted by automatic processing of Earth observation data (Lu et al., 2013; Pesaresi et al., 2016). Furthermore, Different from LandScan, which aims at representing the ambient population, namely the average population over a typical diurnal cycle (Elvidge et al., 2007), GHSL population grids represent the residential population in buildings (Corbane et al., 2017). The building-related statistics in the 2010-census are also for residential buildings. Therefore, the GHSL population grids are the best candidate ancillary information for this

paper to disaggregate the urbanity level building-related statistics extracted from the 2010-census records into grid level. The high correlation ($R^2 = 0.9662$, as shown in Fig. 1) between the GHSL population and the 2010-census recorded population at the county-level further indicates its appropriateness. Detailed county-level population correlation analyses for each of the 31 provinces in mainland China are also provided and can be found from the online supplement. The accesses to the remote sensing data mentioned above are provided in the Data/Code

Availability section.

**2.3 Assign urbanity attribute (urban/township/rural) to the geo-coded grids in the 2015 GHSL population density profile**

In the 2015 GHSL population density profile, the number of populations in each geo-coded grid is given (it is worth noting that this dataset has been updated in 2019 during the preparation of this work). The original resolution

of the 2015 GHSL population density profile is 250m×250m. For computational convenience, it is resampled to 1km×1km resolution before further analysis. Based on the urbanity level residential building-related statistics extracted from the 2010-census records, a top-down dasymetric mapping method will be performed to disaggregate the urbanity level statistics into 1km×1km resolution grids for mainland China. The urbanity attribute of statistics in the 2010-census records is determined according to the administrative unit of the surveyed population. For

example, if a residence is from a village, then the related statistics are aggregated into rural urbanity level; and if from a town, then it is township level; if from a city, it is urban level. However, for the geo-coded population grids in the 2015 GHSL profile, the corresponding urbanity attributes remain to be defined. Therefore, before performing the disaggregation, we will first define the urbanity attribute of each geo-coded grid in the 2015 GHSL profile by applying the reallocation approach developed by Aubrecht and Leon Torres (2015) and illustrated in Gunasekera

et al. (2015).





Aubrecht and Leon Torres (2015) identify the geospatial areas of mixed and residential grids within the urban extent of Cuenca City, Ecuador by using the Impervious Surface Area (ISA) data as they show strong spatial correlations with the built-up areas. The assumption behind their method was that intense lighting is associated with a high likelihood of commercial and/or industrial presence (which is commonly clustered in certain parts of a city, such as central business districts and/or peripheral commercial zones, and such areas are defined as "mixed-use area"), and areas of low light intensity are more likely to be pure residence zone (defined as "residential use area"). In Gunasekera et al. (2015), a similar procedure was used in developing the building stock model for the entire globe. The difference is that Gunasekera et al. (2015) sorted the grids according to the population density in the LandScan population dataset and assigned the gird with urban/rural attributes. For each country, the largest and most populated contiguous grids are classified as urban. This step was repeated iteratively until the urban population proportion for each country was reached.

In this paper, to assign the urbanity attributes (namely urban/township/rural) to geo-coded population grids in the 2015 GHSL profile, for each province we follow the urban/township/rural population proportions (as listed in Table 3) derived from the population statistics in the Short Table of the 2010-census. The assumption behind this urbanity attribute assignment practice is that the larger the population density in a grid, the higher its potential to be assigned as "urban". An example demonstrating the distribution of the 2015 GHSL population grids assigned with urban/township/rural attributes for Baoshan District of Shanghai is shown in Fig. 2. For instance, in Shanghai, the urban/township/rural population proportion derived from the 2010-census records is 76.64%, 12.66%, and 10.7%, respectively. Then, following Gunasekera et al. (2015), the grids (1km×1km) in the 2015 GHSL profile of Shanghai are sorted from the largest to the smallest in population density. The population in those most populated grids are selected and summed up until the urban population proportion (i.e., 76.64% for Shanghai) is reached. Then those selected grids are assigned with the "urban" attribute and the smallest population among these grids determines the threshold to divide urban and non-urban grids (for Shanghai this urban/non-urban grid population threshold is **4936** per km$^2$). For the remaining non-urban grids, the same process is repeated iteratively until the township population proportion (i.e., 12.66% for Shanghai) is reached. These grids are assigned with the "township" attribute and the smallest population among these grids determines the threshold to divide township and rural grids (for Shanghai this township/rural grid population threshold is **2750** per km$^2$). The remaining grids are thus assigned with the "rural" attribute. The urban/township and township/rural population thresholds for 31 provinces in mainland China are listed in Table 3. This process is repeated for all provinces.

**2.4 Residential building stock modeling process**

The following section will introduce the key steps in residential building stock modeling, including the disaggregation of urbanity level statistics extracted from the 2010-census records into grid level, the reclassification of building subtypes with both structure type and storey class, the derivation of residential building floor area and replacement value in each grid. The flowchart in Fig. 3 gives an overview of the whole modeling process.

**2.4.1 Step 1 - Disaggregate urbanity level building-related statistics from the 2010-census into grid level**

Like in many other countries, the population and housing census data in mainland China are particularly surveyed for residential buildings. Therefore, the building stock model developed in this paper is for residential building




stock. As listed in Table 2, building-related statistics extracted from the 2010-census records include the number
of families living in buildings grouped either by the number of the storey (i.e., 1, 2-3, 4-6, 7-9, ≥10) or by structure
type (i.e., steel/reinforced-concrete, mixed, brick/wood, other; hereafter steel/reinforced-concrete is abbreviated
as steel/RC; and "mixed" refer to different combinations of masonry buildings), the average population per family
and the average floor area per capita. For each urbanity level of each province, **the number of families** living in
buildings grouped by storey number or structure type is extracted from the Long Table of the 2010-census, which
is based on the survey of only 10% of the total population in mainland China (as noted in Table 1). Therefore, the
number of families living in different building types needs to be extended from 10% to 100% population first. This
is achieved directly by multiplying the number of families with the factor of 10 (namely factor $F0$ in Step 1-1 of
Fig. 3). Multiplying the number of families with the average number of population per family (namely factor $F1$
in Step 1-2 of Fig. 3, with values listed in Table 2) provides **the number of populations** living in buildings grouped
by storey number (1, 2-3, 4-6, 7-9, ≥10) or structure type (steel/RC, mixed, other, brick/wood) for each urbanity
of each province.

The geo-coded population grids in the 2015 GHSL profile with assigned urbanity attributes (Sect. 2.3) and the
number of populations living in buildings grouped by storey number or structure type derived for each urbanity of
each province seem to allow the direct disaggregation of the 2010-census statistics into the 2015 GHSL grids.
However, the GHSL population is for the year 2015, while the derived population living in different structure type
or storey class from the building-related statistics is for the year 2010. The increase in population/building from
2010 to 2015 must be considered. Here we assume that the increase in population living in buildings grouped by
storey class or structure type from 2010 to 2015 is equal to the increase in population from the 2010-census records
to the 2015 GHSL profile. Therefore, for each urbanity of each province, the derived number of populations living
in building types grouped by storey class or structure type (after performing Step 1-1 and 1-2 in Fig. 3) will be
further amplified to the year 2015 by multiplying the population amplification factor (namely factor $F2$ in Step 1-
3 of Fig. 3). For each urbanity of each province, the value of $F2$ is equal to the ratio of the 2015 GHSL population
to the sum of the population living in buildings of different occupancy types. For example, in urbanity "1001" of
Anhui province in Table 2, the value of $F2$ (1.32) results from the ratio of the 2015 GHSL population (12165295)
to the product of the number of families living in three occupancy types (331730+9035+287 = 341052; based on
surveys of 10% of the whole population), the average number of population per family ($F1$ = 2.71), and the factor
to extend the 10% population survey to 100% population ($F0$ = 10), namely 12165295 / (341052×2.71×10) =
1.32.

Thus, for each urbanity of each province, the number of populations living in buildings grouped by storey class or
structure type in 2015 is derived by multiplying the original number of families living in different building types
(based on surveys of 10% of the whole population) in Table 2 with factor $F0$, $F1$, $F2$. These urbanity level statistics
can be disaggregated into the geo-coded grids of the 2015 GHSL profile. The population share in each grid (relative
to the sum of population of grids with the same urbanity) is used as the disaggregation weight (namely factor $F3$
in Step 1-4 of Fig. 3). By multiplying the urbanity level population living in buildings grouped by storey class or
structure type with the disaggregation factor $F3$ of each grid, the grid level number of populations living in
buildings grouped by storey class or structure type can be directly derived.





### 2.4.2 Step 2 - Derive the population living in the 17 building subtypes within each grid

As explained in Section 2.4.1, after multiplying the original number of families living in different building types extracted from the 2010-census records (Table 2, based on surveys of 10% of the whole population) with factor
*F0*, *F1*, *F2,* and *F3* in Step 1 of Fig. 3, the grid level populations living in buildings grouped either by the number of storey (1, 2-3, 4-6, 7-9, ≥10) or by structure type (steel/RC, mixed, other, brick/wood) are derived for all geo-coded grids in the 2015 year level. To further estimate the residential building floor area and replacement value in each grid, we need to evaluate the unit construction prices of the building types in each grid. Currently, the building types are grouped either by storey number or by structure type, and they need to be reclassified into building
subtypes with both storey class and structure type attributes. Then it will be easier and more reasonable to estimate the unit construction prices of these building subtypes, compared to the estimation made in studies based on building occupancy type (e.g., Wu et al., 2019).

In the following description, we will first introduce the reclassification of building subtypes with both storey class and structure type attributes. Then we will estimate the population living in each of the 17 building subtypes. Based
on the statistics of average floor area per capita in each urbanity level extracted from the 2010-census records (as listed in Table 2), the total floor area of each of the 17 building subtypes in each grid can be derived. Finally, for each building subtype, their replacement value emerges from a multiplication of the floor area with the unit construction price.

By combining the five storey classes (1, 2-3, 4-6, 7-9, ≥10) with the four structure types (steel/RC, mixed, other,
brick/wood), the building types in the 2010-census records can be initially reclassified into 20 building subtypes. According to Hu et al. (2015) and Wang et al. (2018), most brick/wood buildings are with quite low height (1 or 2-3 storey), while steel/RC buildings are generally quite high with 10-storey height and above. Therefore, in this paper it is assumed that for "brick/wood" structure type, there are only two storey classes (1, 2-3); while for "steel/RC", "mixed", and "other" structure types, all five storey classes (1, 2-3, 4-6, 7-9, ≥10) are available (namely
the assumptions in Step 2-1 and 2-2 of Fig. 3). Thus, the number of building subtypes with known storey class and structure type is reduced from 20 to 17. The abbreviations of these 17 building subtypes are listed in Table 4.

After performing the calculations in Step 1 of Fig. 3, the grid level populations living in buildings grouped either by the number of storey (1, 2-3, 4-6, 7-9, ≥10) or by structure type (steel/RC, mixed, other, brick/wood) are derived for all geo-coded grids. Thus, we know in each grid the number of population living in buildings of the
five storey classes, but we do not know for each storey class how the population is distributed among the four structure types. Also, we know how many people live in steel/RC buildings or other structure types, but for each structure type, we do not know how they are distributed into the five storey classes. For each grid, to derive the number of population living in each of the 17 building subtypes with known structure type and storey class, we need to solve 17 unknown variables from 9 equations. The 9 equations are listed as follows:

$$BRIWOMC1 + STLRCMC1 + MIXEDMC1 + OTHERMC1 = Num_{storey1} \quad (1)$$
$$BRIWOMC23 + STLRCMC23 + MIXEDMC23 + OTHERMC23 = Num_{storey23} \quad (2)$$
$$STLRCMC46 + MIXEDMC46 + OTHERMC46 = Num_{storey46} \quad (3)$$
$$STLRCMC79 + MIXEDMC79 + OTHERMC79 = Num_{storey79} \quad (4)$$
$$STLRCMC10 + MIXEDMC10 + OTHERMC10 = Num_{storey10} \quad (5)$$
$$BRIWOMC1 + BRIWOMC23 = Num_{BRIWO} \quad (6)$$





$$STLRCMC1 + STLRCMC23 + STLRCMC46 + STLRCMC79 + STLRCMC10 = Num_{STLRC} \quad (7)$$
$$MIXEDMC1 + MIXEDMC23 + MIXEDMC46 + MIXEDMC79 + MIXEDMC10 = Num_{MIXED} \quad (8)$$
$$OTHERMC1 + OTHERMC23 + OTHERMC46 + OTHERMC79 + OTHERMC10 = Num_{OTHER} \quad (9)$$

The 17 to-be-solved variables on the left side of this equation set represent the numbers of populations living in

the 17 buildings subtypes (as defined in Table 4); on the right side, the numbers are populations living in buildings classified by fives storey class and four structure types, which are already known after performing the calculations in Step 1 of Fig. 3. Since this set of 9 equations contains 17 unknown variables, it is an underdetermined linear problem. In order to provide values for the 17 unknowns, additional assumptions have to be utilized.

The strategy we employ here to derive the population living in each of the 17 building subtypes of each grid is a

series of distribution steps based on a prioritized ranking of building types and storey classes. For example, we first assign 1 storey class buildings into brick/wood structure type and distribute≥10-storey class as steel/RC structure type (following the assumptions in Step 2-1 and 2-2 of Fig. 3). Although this distribution strategy may deviate from the actual situation, the basic requirement, that in each grid the sum of the population living in the 17 building subtypes is equal to the population living in building types grouped by structure type or by storey class,

is satisfied. The main distribution steps are summarized in Appendix A.

### 2.4.3 Step 3 - Derive the residential floor area of the 17 residential building subtypes in each grid

Based on the distribution processes in Appendix A, we derive the number of populations living in each of the 17 building subtypes in each gird. To derive the residential floor area of each building subtype, the average residential floor area per capita is needed, which is given in the Short Table of 2010-census (namely factor *F4* in Step 3-1 of

Fig. 3) for each urbanity level of each province. Therefore, the floor area of the 17 building subtypes in each grid can be directly derived. This grid level residential building floor area distribution map is available from the online supplement. Comparison between the modeled floor area and the 2010-census recorded floor area for residential buildings at county/district-level will be performed in Sect. 3.2.2.

### 2.4.4 Step 4 - Derive the replacement value of the 17 residential building subtypes in each grid

With the residential building floor area for each building subtype in each grid being derived in Step 3, to get the corresponding replacement value, the unit construction prices of the 17 building subtypes need to be estimated (namely factor *F5* in Step 4-1 of Fig. 3). Given the uniqueness of the building reclassification strategy adopted in this paper, there are no standard unit construction price evaluations for the building subtypes we use here. Therefore, we estimate the unit construction prices of the 17 building subtypes (as listed in Table 4) by averaging

the construction prices given in different literature (e.g., 2015 China Construction Statistical Yearbook, the World Housing Encyclopedia, real-estate agency reports, etc.). For the 17 building subtypes in each grid, by multiplying their floor area with the corresponding unit construction price in Table 4, their replacement values can be directly derived. This grid level residential building replacement value distribution map is also available from the online supplement. We emphasize that in this paper, the term "replacement value" refers to the amount of money needed

to rebuild a property exactly as it is before its destruction regardless of any depreciation, namely the gross capital stock. A prefecture-level comparison between our modeled residential building replacement value and the wealth capital stock value in Wu et al. (2014) will be given in Sect. 3.2.1.



## 3. Results and Performance Evaluation

### 3.1 Results

**3.1.1 Modeled floor area and replacement value for residential buildings in each urbanity of each province**

The grid level residential building floor area and replacement value (unit: RMB, in 2015 current price) are aggregated into urbanity level (urban/township/rural) for each province, as listed in Table 5. The total modeled residential building floor area for mainland China in 2015 reaches 42.31 billion $m^2$. By applying the same unit construction prices for the same 17 building subtypes in all the urban/township/rural areas of the 31 provinces, the

initially modeled replacement value of residential buildings in mainland China is 77.8 trillion RMB (in 2015 current price). It is clear that like all other building stocks, the Chinese building stock is a complicated economic, physical and social system (Yang and Kohler, 2008). There are significant differences across the country in terms of economic development level, geographic climatic diversity, and standardization in building construction. Therefore, it is mainly for computational convenience that this paper applies the same unit construction price for

all the provinces and all the urbanity levels. To improve accuracy in future seismic risk assessment, the unit construction prices of specific building types in the target study area should be adjusted accordingly.

### 3.1.2 An example illustrating the distribution of modeled floor area in Shanghai

For better visualization of the modeled floor area at grid level, we plot the residential building floor area distribution map and the 2015 GHSL population of Shanghai as an example. As can be seen from Fig. 4, grids

with a high density of floor area typically cluster in the downtown area (including eight administrative districts, namely Yangpu, Hongkou, Zhabei, Putuo, Changning, Xuhui, Jing'an, and Huangpu) and the Pudong district. This corresponds to the fact that these districts are the most developed in Shanghai. As revealed by the 3D-view of the population distribution in panel (c) of Fig. 4, districts with a high density of floor area also have a high population density.

### 3.2 Performance Evaluation

As of now, we have developed a high-resolution (1km×1km) residential building stock model (in terms of floor area and replacement value) for mainland China. This model is established by disaggregating the urbanity level building-related statistics in 2010-census records into grid level and using the 2015 GHSL geo-coded population as the disaggregation weight. Due to the approximations and assumptions made in the modeling process, the

reasonability and consistency of the modeled results need to be evaluated. Due to the typical lack of official statistics on high-resolution building stock from the government (Wu et al., 2018), direct comparison of the modeled floor area and replacement value at grid level with that from official census or statistical yearbooks are not instantly available. Instead, we will compare our modeled results with other studies or census records at a coarser level. Moreover, since the development of such a high-resolution residential building model is mainly

targeted for seismic risk assessment in mainland China, we will also apply our modeled results to seismic loss estimation combining with the 2008 Wenchuan Ms8.0 earthquake intensity map and an empirical loss function. The estimated losses will be compared with those recorded in affected counties/districts of Sichuan Province.



### 3.2.1 Prefecture-level comparison between the modeled residential building replacement value and the net capital stock value estimated in Wu et al. (2014)

Due to the lack of officially published datasets on the value of fixed capital stock in China (Wu et al., 2018), previous studies (e.g., Holz, 2006; Wang and Szirmai, 2012) mainly employed the perpetual inventory method (PIM) in which economic indicators (e.g., gross fixed capital formation, total investment in fixed assets, etc.) are used. The resolutions of these estimations were almost exclusively limited at national/provincial-level (Wu et al., 2014). This coarse spatial resolution forms a major obstacle in applying the model in disaster loss estimation, where high-resolution hazard data are used. To overcome this gap, Wu et al., (2014) estimated the net capital stock values from 1978 to 2012 for 344 prefectures in mainland China by using the PIM. In their Appendix Table A1, the net capital stock values calculated in 2012 current price for 344 prefectures were provided, with the depreciation of all exposed assets (i.e., residential and non-residential building structures, tools, machinery, equipment, and infrastructure) being considered.

To compare with the net capital stock value in Wu et al. (2014), the grid level residential building replacement value modeled in this paper (namely the gross value of residential building stock) was aggregated into prefecture-level. Pearson's correlation coefficient ($R^2$) was used to measure the degree of collinearity between two datasets, with higher $R^2$ indicating a stronger correlation. As shown in Fig. 5, there is a high correlation ($R^2 = 0.9512$) between our residential building replacement values and the net capital stock values in Wu et al. (2014) at the prefecture-level. The absolute replacement value of residential buildings is around 0.54 times the net capital stock value in Wu et al. (2014). To explain this discrepancy, we collected the annual fixed asset investment on residential buildings and on all types of buildings for each of the 31 provinces during the years 2004-2014 from the statistical yearbooks (detailed statistics are available from the online supplement). As can be seen from Fig. 6, for each province the sum of fixed asset investment on residential buildings during 2004-2014 is around 0.45 times the investment on all types of buildings, quite close to the 0.54 ratio in Fig. 5. The replacement value we estimate is purely for residential buildings without depreciation, while the net capital stock value in Wu et al. (2014) includes depreciation of all exposed assets (residential, non-residential buildings, infrastructures, and equipment). Thus, we consider our model results as reasonable.

### 3.2.2 County/prefecture-level comparison between modeled residential building floor area and records in the 2010-census

    Compared with previous studies related to building stock modeling in China, we have used finer urbanity level building-related statistics as input to generate the grid level residential building stock model. In each urbanity, the building-related statistics extracted from the 2010-census records are from areas with a similar development background, but they belong to different administrative units (i.e., prefectures and counties). Also, within the same prefecture or county, the geo-coded grids are of different urbanity attributes. Therefore, the reliability of our model can be better proved if the modeled results correlate well with actual records at the county or prefecture-level. After a thorough search, we find that county-level records of residential building floor area are also available for 28 provinces in mainland China, except for Hunan, Liaoning, and Sichuan provinces, for which only prefecture-level records of residential building floor area can be found from the 2010-census records. Then, to compare our modeled floor area with the 2010-census records at the county/prefecture-level, the modelled grid level residential

building floor area was first aggregated into counties/districts for the 28 provinces, and prefectures for Hunan, Liaoning, and Sichuan, respectively. The final comparison between our estimated residential building floor area with that recorded in the 2010-census is plotted in Fig. 7.

As can be seen from Fig. 7, there is a high correlation ($R^2 = 0.9376$) between modeled floor area and that recorded in the 2010-census at the county/prefecture-level. The regression relation indicates that our modeled floor area for 2015 is around 1.14 times that in the 2010-census. In Step 1-3 of the modeling process (Fig. 3), for each urbanity level of each province, the building-related statistics extracted from the 2010-census records were amplified into the 2015 level by multiplying the factor $F2$. Mathematically speaking, $F2$ is the ratio of the 2015 GHSL population to the 2010-census recorded population. $F2$ is 1.13 for the whole mainland China, which can be derived by

following the derivation process of $F2$ illustrated in Sect. 2.4.1 based on the statistics in Table 2. Therefore, we consider the ratio of 1.14 between our modeled floor area for 2015 and that recorded in the 2010-census at the county/prefecture-level as quite reasonable. For each province, we also plotted the correlation analyses for the population (between the 2015 GHSL population and 2010-census recorded population) and for the residential building floor area (between the modeled floor area and the 2010-census recorded floor area), which are available

from the online supplement. The corresponding regression parameters and correlation coefficients for the population and the residential building floor area of each province are listed in Table 6.

From Table 6 we can see that the correlation between the 2015 GHSL population and the 2010-census recorded population, and the correlation between the modeled floor area and the 2010-census recorded floor area are generally very high for a majority of provinces (with $R^2 \geqslant 0.9$). This indicates the plausibility of choosing the

2015 GHSL population as the ancillary information to disaggregate the urbanity level building-related statistics, and the reliability of our modeled floor area at the county/prefecture-level. However, it is also worth noting that for coastal provinces like Fujian and Jiangsu, the correlation coefficients of floor area are lower (with $R^2 < 0.82$). We explain this discrepancy with an overpredicted population in the 2015 GHSL profile for the capital or the most developed cities in these provinces (as can be checked from the population correlation analyses for these provinces

from the online supplement). Many people tend to work in the capital or the most developed cities without being officially registered as residents. These people are not counted in the 2010-census of these cities but are included in the 2015 GHSL population density profile, which is derived from remote sensing data combined with the actual population density.

**3.2.3 Application of the residential building stock model to seismic loss estimation**

Since the residential building model developed in this paper is targeted for seismic risk analysis, we now use the modeled replacement value to estimate the seismic loss to residential buildings in Sichuan province caused by the Wenchuan Ms8.0 earthquake. The hazard component used for this loss estimation is the macro-seismic intensity map of the 2008 Wenchuan Ms8.0 earthquake (Fig. 8), which was issued by the China Earthquake Administration (CEA) based on post-earthquake field investigations. The vulnerability function used was the empirical loss

function developed in Daniell (2014, Page 242) for mainland China, which provides the relation between macro-seismic intensity and loss ratio (the ratio between repairment cost and replacement cost of buildings damaged in an earthquake). This empirical vulnerability function was developed based on reported seismic damage and loss related to earthquakes that occurred in mainland China in the past few decades. Such information was retrieved



through an extensive collection of damage and loss records from journals, books, reports, conference proceedings,
and even newspapers.

Our estimated seismic loss of residential buildings in Sichuan province due to the Wenchuan Ms8.0 earthquake is
around 432 billion RMB (in 2015 current price). The spatial distribution of loss ratios, i.e., the ratio of the estimated
loss to the total residential building replacement value in counties/districts of Sichuan province, is shown in Fig.
9. In other reports and studies on the loss assessment of the Wenchuan earthquake, e.g., in Yuan (2008), the
estimated loss to residential buildings in Sichuan province was around 170 billion RMB (in 2008 current price).
The officially issued loss estimated by the Expert Panel of Earthquake Resistance and Disaster Relief (EPERDR,
2008) to residential buildings in Sichuan province was around 98.3-435.4 billion RMB, with the median loss
around 212.32-247.25 billion RMB (in 2008 current price). It should be noted that in these studies, the unit
construction price used for rural/urban/township building replacement was around 800-1500 RMB per m$^2$, which
is 1/2.5-1/1.5 of the unit construction price used in this paper as listed in Table 4. Dividing our estimated loss by
the factor of 1.5-2.5, then the difference in construction price used in this paper and previous studies are eliminated,
and the estimated loss based on our building exposure model turns from 432 billion to around 144-288 billion
RMB (in 2015 current price), which is now consistent with that estimated by EPERDR and Yuan (2008). This
simple test further indicates the applicability of our model in seismic loss estimation. Thus, the grid level residential
building floor area and replacement value developed in this paper can be regarded as reliable exposure inputs for
future seismic risk assessment in mainland China.

## 4.    Limitations in the model and directions for future improvement

According to studies on assessing the resolution of exposure data required for different types of natural hazards
(e.g., Chen et al., 2004; Thieken et al., 2006; Figueiredo and Martina, 2016; Röthlisberger et al., 2018), the
1km×1km residential building stock model developed in this paper is sufficient for seismic risk assessment.
However, limitations in our model are inevitable due to the assumptions and approximations employed in the
modeling process. For example, when disaggregating the urbanity level building-related statistics in the 2010-
census into grid level and scaling these statistics from 2010 to 2015, we assume that the number of residential
buildings in each grid is proportional to its population weight and the increase in building-related statistics of each
urbanity is equal to its population increase, which needs to be carefully evaluated by the local development of
building stock (e.g., Fuchs et al., 2015). Secondly, to derive the population living in each of the 17 building
subtypes in each grid, we assume that brick/wood buildings are limited to 1 and 2-3 storey classes and distribute
the number of steel/RC buildings to ≥10-storey class first, which may not be fully consistent with the real cases.
Furthermore, we use the same unit construction prices for the same building subtypes regardless of their variation
across province and urbanity, which also needs certain readjustment when applying our modeled residential
building replacement value into actual seismic risk analyses.

In the future, with the increasing availability of open source datasets that track individual building features in detail,
the current limitations in this paper can possibly be overcome. Attempts have been made to combine publicly
available building vector data (which contains the spatial location, footprint, and height of each building) and
census records to improve the exposure estimation (e.g., Figueiredo and Martina, 2016, Wu et al., 2019, Paprotny
et al., 2020). Algorithms to extract building footprints and height from aerial imagery and using computer vision





techniques have been used by commercial companies like Google and Microsoft (Parikh, 2012; Bing Maps Team, 2014). More recently, by using an unmanned aerial vehicle and a convolutional neural network, Xiong et al. (2020) introduced an automated building seismic damage assessment method, in which not only the 3D building structure can be constructed, but also the building damage state can be predicted automatically with an accuracy of 89%. We take these attempts as an indicator that the high-resolution modeling of building stock for individual buildings will become more widely available in the future.

## 5. Conclusion

In this paper, a 1km×1km resolution residential building stock model (in terms of floor area and replacement value) targeted for seismic risk analysis for mainland China is developed, by using the 2015 GHSL population density profile as the bridge and by disaggregating the finer urbanity level 2010-census records into grid level for each province. In each grid, a building distribution strategy is adopted to derive the number of population living in each of the 17 building subtypes with structure type and storey class attributes, based on which the floor area and replacement value of each building subtype are derived. In each urbanity of each province, the building-related statistics extracted from the 2010-census records are from areas with a similar development background but different administrative units (i.e., prefectures and counties). Therefore, to evaluate the model performance, the residential building replacement value is first compared with the net capital stock value estimated in Wu et al. (2014) at the prefecture-level. These two datasets are well correlated, and the former is around 0.45 of the latter, which is quite reasonable referring to the fact that for each province the sum of fixed asset investment value on residential buildings is around 0.54 of the sum of investment values on all types of buildings during 2004-2014. Furthermore, county/prefecture-level comparisons of the residential floor area modeled in this paper with records from the 2010-census are also conducted. It turns out that the modeled and recorded residential building floor areas are highly compatible for many counties and prefectures. To further check the applicability of the modeled results in seismic risk assessment, an empirical seismic loss estimation is performed based on the intensity map of the 2008 Wenchuan Ms8.0 earthquake, the empirical loss function in Daniell (2014), and our modelled replacement value of residential buildings in Sichuan province. By reducing the difference in unit construction price used in this paper and other studies, our estimated loss range is consistent with the loss derived from damage reports based on field investigation. These comparisons indicate the reliability of the geo-coded grid level residential building exposure model developed in this paper. More importantly, the whole modeling process is fully reproducible, and all the modeled results are available from the online supplement, which can also be easily updated when more recent or detailed census data are available.

## Appendix

In Appendix A, to derive the population living in each of the 17 building subtypes of each grid, the distribution strategy mentioned in Sect. 2.4.2 is explained in detail. In addition, a MATLAB script is provided to help understand this strategy.



## Data/Code Availability

The accesses to data used or mentioned in this paper are as follows: (1) 2010 China Sixth Population Census Tabulation (http://www.stats.gov.cn/tjsj/pcsj/rkpc/6rp/indexch.htm); (2) 2015 Global Human Settlement Layer (GHSL) population density profile (http://data.europa.eu/89h/jrc-GHS-ghs_pop_gpw4_globe_r2015a) ; (3) The spatial

645 administrative boundaries from the National Geomatics Centre of China (http://www.ngcc.cn/ngcc/html/1/391/392/16114.html); (4) The Globcover land cover maps (http://due.esrin.esa.int/page_globcover.php); (5) The GLC2000 landcover classes (https://forobs.jrc.ec.europa.eu/products/glc2000/legend.php); (6) The MODIS imaging project (https://modis.gsfc.nasa.gov/about/); (7) The GlobeLand30 project (http://www.globallandcover.com/); (8) The DMSP-OLS nighttime light datasets (https://data.noaa.gov/metaview/page?xml=NOAA/NESDIS/NGDC/STP/DMSP/iso/xml/G01119.xml&view=getDataView&header=none); (9) OpenStreetMap (https://www.openstreetmap.org/); (10) Gridded Population of the World (GPW, http://sedac.ciesin.columbia.edu/gpw/global.jsp); (11) Global Rural-Urban Mapping Project-Population (GRUMP-population, https://sedac.ciesin.columbia.edu/data/collection/grump-v1); (12) LandScan Global Population Datasets (https://landscan.ornl.gov/landscan-datasets); (13) WorldPop/AsianPop (https://www.worldpop.org/geodata/listing?id=29); (14) PopGrid China (http://www.geodoi.ac.cn/edoi.aspx?DOI=10.3974/geodb.2014.01.06.V1); (15) An example illustrating the multi-variate equation solving process in Sect. 2.4.2, including the input file and the MATLAB script that are available from the online supplement.

## Supplement

The supplementary data related to this work are available online at https://doi.org/10.5281/zenodo.4669800.

## Author contribution

DX conducted the data collection and preparation, analyses of results, model validation, and prepared the draft manuscript. JD guided the data collection and preparation process, developed the modeling methodology and performed the calculation and co-analysed the results. HT and FW supervised the project and provided advice and feedback in the process. All authors contributed to the revision of the manuscript.

## Competing interests

The authors declare that they have no conflict of interests.

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

**Figures**

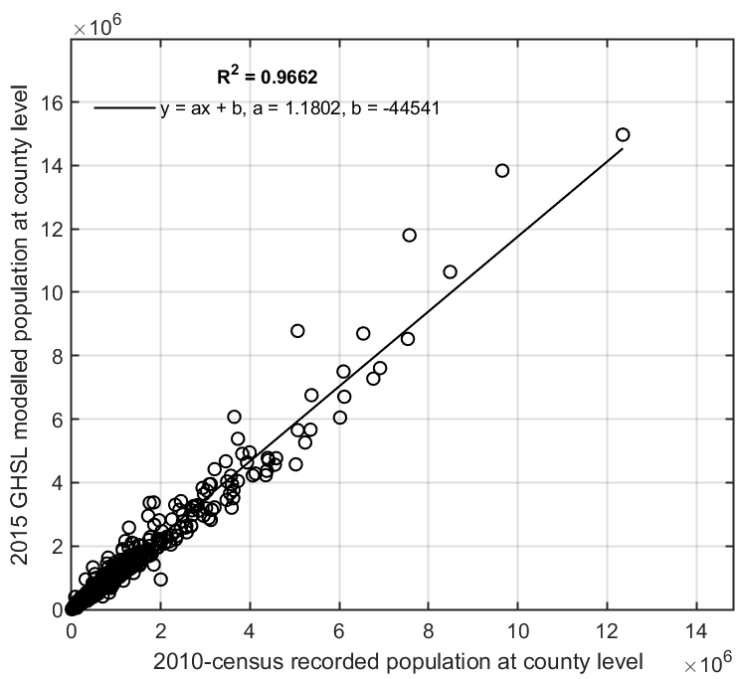

**Figure 1**: County-level comparison of the population between the 2015 GHSL profile and the 2010-census records.

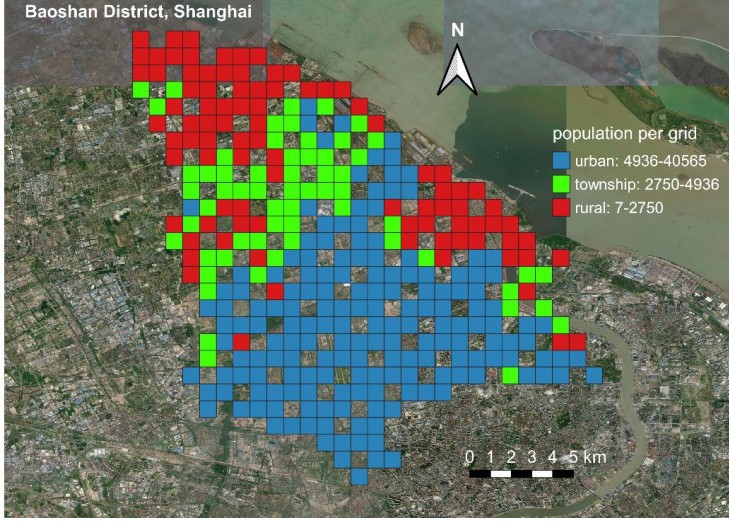

**Figure 2**: An example showing the assignment of urbanity attribute in the 2015 GHSL population grids for Baoshan district in Shanghai. The urban/township and township/rural population thresholds for Shanghai are 4936/km$^2$ and 2750/km$^2$, respectively (see context in **Sect. 2.3** for more details). This figure is plotted by using the QGIS platform (https://qgis.org/en/site/) and the background satellite map is provided by Bing map service (© Microsoft).





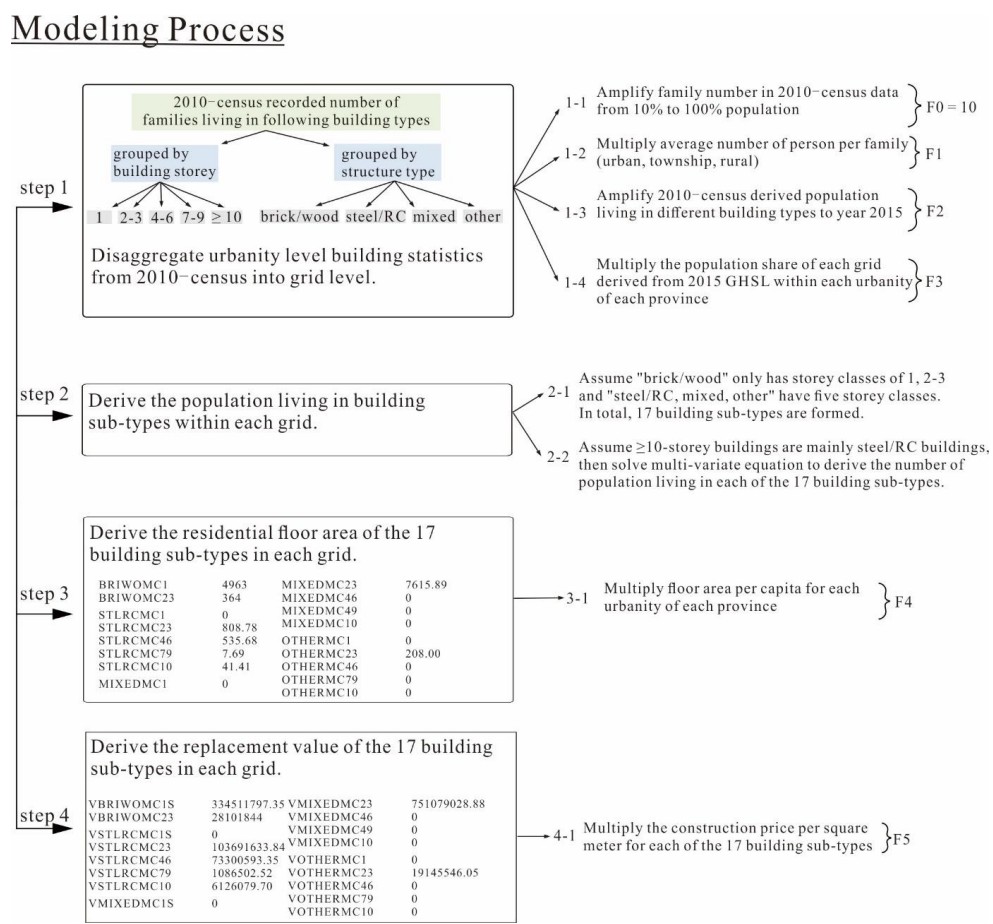


**Figure 3**: Flowchart of the residential building stock modeling process adopted in this paper (see context in **Sect. 2.4** for more details).


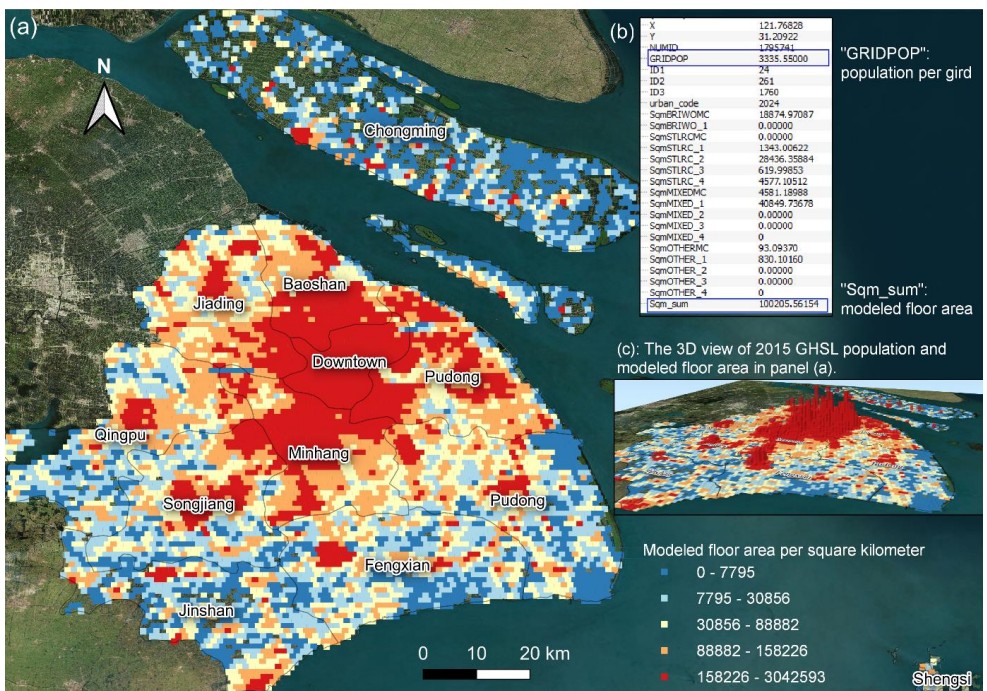

**Figure 4**. An example illustrating the building stock model of Shanghai: (a) The distribution of modeled floor area
(unit: $m^2$) in each 1km×1km grid; (b) A table showing the modeled floor area of the 17 building subtypes, the total
population "GRIDPOP" and the total modeled floor area "Sqm_sum" in an example grid; (c) The 3D view of the
modeled floor area and the 2015 GHSL population (the height of the cuboid in each grid is proportional to its
population density). This figure is plotted by using the QGIS platform and the background satellite map is provided
by the Bing map service (© Microsoft).



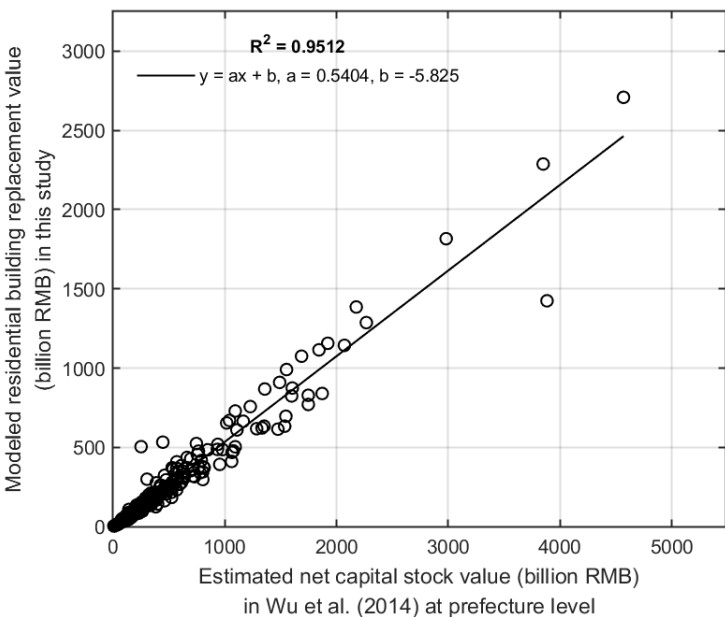


**Figure 5**: Prefecture-level comparison of the modeled residential building replacement value in this paper (unit: billion RMB in 2015 current price) with the net capital stock value estimated in Wu et al. (2014) by using the perpetual inventory method (unit: billion RMB in 2012 current price). Note: the net capital stock value estimated in Wu et al. (2014) includes the depreciated value of all exposed elements, namely the residential buildings, non-875 residential buildings, infrastructures, and equipment (see context in **Sect. 3.2.1** for more details).





**Figure 6**: Comparison of the sum of the annual fixed asset investment (unit: billion RMB) on residential buildings with investment on all types of buildings during 2004-2014 in each of the 31 provinces in mainland China. Detailed investment statistics are available from the online supplement.

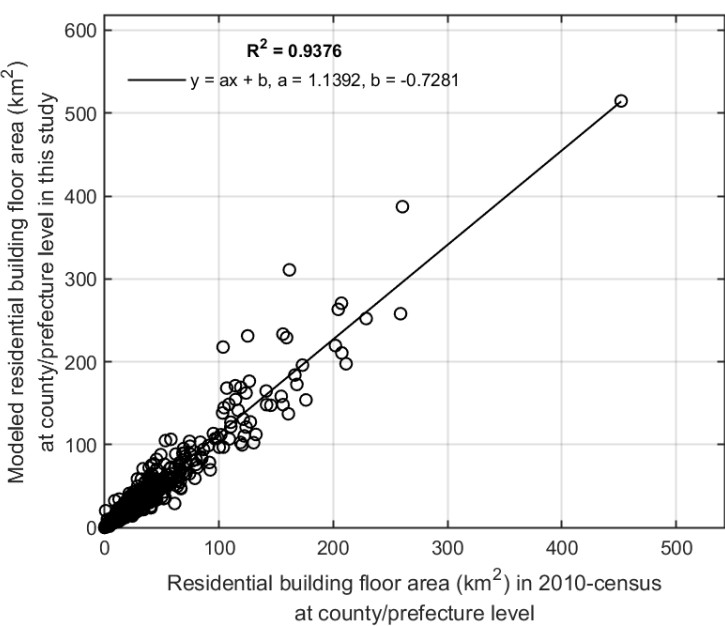


**Figure 7**: County/prefecture-level comparison of the modeled residential building floor area (km$^2$) in this paper with that recorded in the 2010-census for 31 provinces in mainland China (see context in **Sect. 3.2.2** for more details).

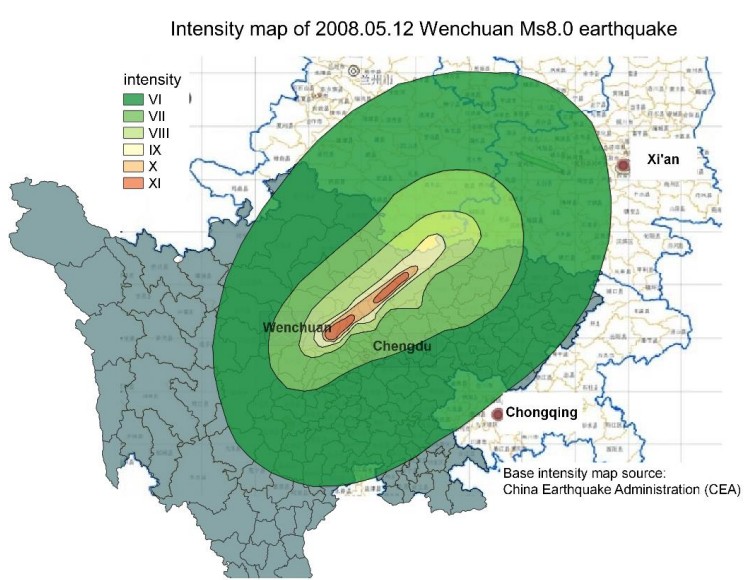

**Figure 8**. Macro-seismic intensity map of the 2008 Wenchuan Ms8.0 earthquake, modified after the base intensity map issued by China Earthquake Administration (CEA).

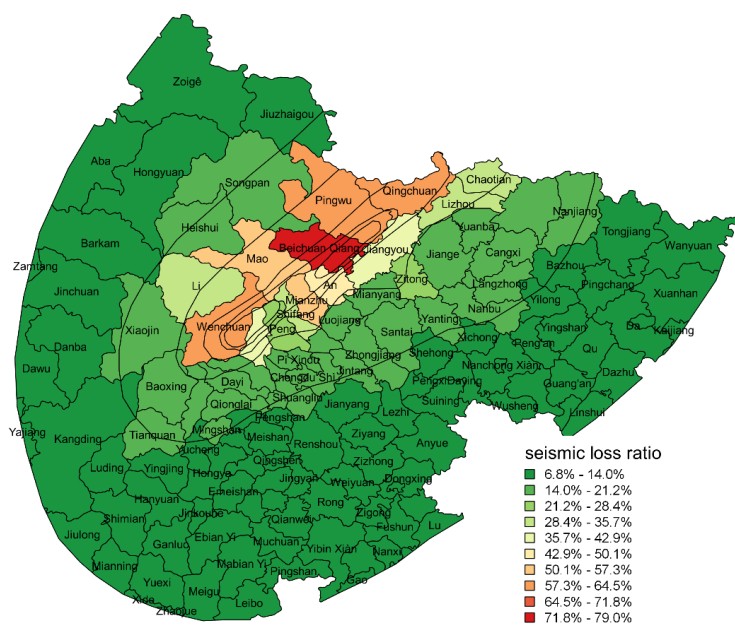

**Figure 9**. Distribution of seismic loss ratio (the ratio between repairment cost and replacement cost) of residential buildings in affected districts/counties of Sichuan province due to the 2008 Wenchuan Ms8.0 earthquake. Black contours represent the extent of each intensity zone of the Wenchuan earthquake (see context in **Sect. 3.2.3** for more details).



## Tables

**Table 1**: Main data sources used in this paper. Accesses to these data are provided in the Data/Code Availability section.

| Data source | Data description | Resolution | Data location | Indicator in this paper | Notes |
|---|---|---|---|---|---|
| 2010-census Short Table | Overall population | urban/township/rural level for each of the 31 provinces in mainland China; (the urbanity level in the census is defined according to the administrative unit of the surveyed population) | Table 1-1a, 1-1b, 1-1c | N/A | Based on surveys of 100% of the population in mainland China |
| 2010 -census Long Table | Number of families living in buildings grouped by usage (residential, commercial, mixed) | | Table 9-1a, 9-1b, 9-1c | N/A | Based on surveys of 10% of the overall population in mainland China |
| | Number of families dwelled in buildings grouped by storey number (1, 2-3, 4-6, 7-9, ≥10) | | | | |
| | Number of families dwelled in buildings grouped by structuretype (steel/RC, mixed, other, brick/wood) | | | | |
| 2010-census Short Table | Average population per family | | Table 1-1a, 1-1b, 1-1c | F2 of Fig. 3 | Based on surveys of 100% of the population in mainland China |
| | Average residential floor area (m$^2$) per person | | Table 1-14a, 1-14b, 1-14c | F4 of Fig. 3 | |
| 2015 GHSL population density profile | The number of populations in each geo-coded grid | 1km×1km | N/A | $\lambda$ | The original resolution is 250m×250m and was resampled to 1km×1km |
| Wu et al. (2014) | The estimated net wealth capital stock value in 344 prefectures of mainland China | Prefecture-level | N/A | N/A | All exposed assets (residential and non-residential buildings, infrastructures, instruments, etc.) and their depreciation are considered |
| 2010-census Short Table | The residential building floor area statistics in administrative units | Prefecture-level for Hunan, Liaoning, and Sichuan; county-level for other 28 provinces | Table 1-1, 1-14 in the 2010-census book of each province | N/A | Some data are downloaded from the commercial website (https://www.yearbookchina.com/) |

Note: The "2010-census" in **Data source** is the abbreviation of the "2010 Population Census of the People's Republic of China"; "**Data location**" refers to the serial number of the table in the original data source (see context in **Sect. 2.1** for more details).



**Table 2:** In each urbanity, the population sum of the 2015 GHSL profile and the residential building-related statistics extracted from the 2010-census records.

| "Urbanity"+"0"+"Prov_ID" | Province | 2015 GHSL population in each urbanity | Floor area per capita (m²) | Aver. pop. per family | Number of families grouped by occupancy | | | Number of families grouped by storey class | | | | | Number of families grouped by structure type | | | | Amp. factor |
|---|---|---|---|---|---|---|---|---|---|---|---|---|---|---|---|---|---|
| | | | | | living | commercial | mixed | 1 | 2-3 | 4-6 | 7-9 | ≥10 | steel/RC | mixed masonry | brick/wood | others | |
| | | | | | | | | **urban** | | | | | | | | | |
| 1001 | Anhui | 12165295 | 29.42 | 2.71 | 331730 | 9035 | 287 | 44093 | 82489 | 175486 | 20922 | 17775 | 135377 | 176462 | 26705 | 2221 | 1.32 |
| 1002 | Beijing | 18598941 | 27.81 | 2.40 | 517975 | 6482 | 988 | 127740 | 193270 | 148238 | 21919 | 22367 | 226367 | 212873 | 83192 | 2025 | 1.47 |
| 1003 | Chongqing | 8402588 | 29.77 | 2.65 | 258417 | 3956 | 247 | 17185 | 39448 | 85383 | 81270 | 126870 | 131656 | 112494 | 13433 | 4790 | 1.21 |
| 1004 | Fujian | 12702780 | 30.29 | 2.70 | 360721 | 13488 | 736 | 30557 | 97680 | 135725 | 79915 | 30332 | 213350 | 124702 | 23948 | 12209 | 1.25 |
| 1005 | Gansu | 5296224 | 26.69 | 2.68 | 160717 | 3134 | 107 | 24489 | 21076 | 75051 | 9074 | 8731 | 66665 | 15057 | 3398 | 3398 | 1.21 |
| 1006 | Guangdong | 56529958 | 26.37 | 2.63 | 1466895 | 34218 | 513 | 152601 | 299326 | 453172 | 183699 | 183699 | 748196 | 663772 | 12463 | 3398 | 1.43 |
| 1007 | Guangxi | 8484803 | 30.71 | 2.93 | 238044 | 5912 | 264 | 26305 | 58876 | 99335 | 11955 | 8601 | 86601 | 138730 | 12463 | 2354 | 1.19 |
| 1008 | Guizhou | 5475276 | 25.94 | 2.82 | 157713 | 5141 | 19 | 17373 | 38055 | 50766 | 49256 | 78055 | 78055 | 75834 | 7703 | 1262 | 1.19 |
| 1009 | Hainan | 2334559 | 25.42 | 3.17 | 56383 | 1602 | 68 | 9674 | 14288 | 13787 | 13124 | 7112 | 41510 | 10814 | 4948 | 713 | 1.27 |
| 1010 | Hebei | 14837665 | 30.10 | 2.95 | 419978 | 3950 | 96 | 100741 | 42944 | 230919 | 29889 | 19435 | 155581 | 211716 | 54745 | 1886 | 1.19 |
| 1011 | Heilongjiang | 14368585 | 23.72 | 2.58 | 455996 | 6911 | 418 | 122051 | 20020 | 130862 | 173283 | 16691 | 163427 | 188660 | 104208 | 6622 | 1.20 |
| 1012 | Henan | 18535815 | 34.02 | 3.05 | 521036 | 7612 | 215 | 79535 | 122569 | 244091 | 64920 | 17533 | 190648 | 307902 | 28268 | 1830 | 1.15 |
| 1013 | Hubei | 175455444 | 33.22 | 2.82 | 502439 | 12753 | 349 | 40937 | 132838 | 179474 | 126270 | 35653 | 180316 | 298109 | 2847 | 2847 | 1.21 |
| 1014 | Hunan | 12920714 | 33.45 | 2.89 | 358447 | 9813 | 501 | 32935 | 92165 | 160007 | 62887 | 20266 | 132713 | 201615 | 31404 | 2528 | 1.21 |
| 1015 | Jiangsu | 30871919 | 33.86 | 2.81 | 876264 | 14961 | 802 | 129293 | 224580 | 412115 | 65052 | 60185 | 325288 | 469388 | 92721 | 3828 | 1.23 |
| 1016 | Jiangxi | 7845049 | 29.76 | 3.19 | 201690 | 3594 | 201 | 17052 | 13029 | 85663 | 48457 | 7385 | 111658 | 76679 | 15396 | 1551 | 1.20 |
| 1017 | Jilin | 10272119 | 25.21 | 2.62 | 329782 | 4910 | 1777 | 59861 | 149906 | 96067 | 175788 | 108325 | 175788 | 108325 | 48852 | 1727 | 1.17 |
| 1018 | Liaoning | 22179450 | 25.76 | 2.57 | 768884 | 7122 | 843 | 111439 | 280046 | 366106 | 211530 | 58885 | 321935 | 381031 | 71386 | 1654 | 1.11 |
| 1019 | Inner Mongolia | 8313523 | 24.86 | 2.67 | 251738 | 6951 | 631 | 84432 | 133932 | 105902 | 11690 | 3658 | 105902 | 87092 | 61924 | 3771 | 1.20 |
| 1020 | Ningxia | 2222156 | 28.38 | 2.71 | 64336 | 1829 | 29 | 10922 | 7958 | 44770 | 1313 | 1202 | 24606 | 34483 | 6352 | 724 | 1.24 |
| 1021 | Qinghai | 1478166 | 27.77 | 2.74 | 41342 | 1229 | 62 | 4877 | 8035 | 20737 | 6292 | 2630 | 26113 | 2415 | 2415 | 516 | 1.27 |
| 1022 | Shaanxi | 9028318 | 28.81 | 2.70 | 269044 | 4820 | 362 | 33723 | 56478 | 122687 | 37356 | 23620 | 89287 | 173753 | 8694 | 2130 | 1.22 |
| 1023 | Shandong | 28926001 | 32.41 | 2.80 | 855282 | 15616 | 242 | 252471 | 88326 | 432226 | 67205 | 30670 | 348873 | 356038 | 161295 | 4692 | 1.19 |
| 1024 | Shanghai | 20564236 | 25.11 | 2.52 | 604654 | 9991 | 928 | 116799 | 304794 | 304794 | 27780 | 104766 | 249438 | 249438 | 93734 | 3096 | 1.33 |
| 1025 | Shanxi | 9838476 | 25.77 | 2.88 | 282847 | 4319 | 87 | 53815 | 47879 | 157087 | 18683 | 9702 | 90187 | 163209 | 29124 | 4646 | 1.19 |
| 1026 | Sichuan | 15739421 | 30.70 | 2.67 | 499024 | 9628 | 630 | 47158 | 79975 | 198299 | 136824 | 46396 | 218827 | 247875 | 34088 | 7862 | 1.16 |
| 1027 | Tianjin | 10012784 | 25.51 | 2.65 | 237060 | 2606 | 167 | 34902 | 12083 | 143755 | 28570 | 20356 | 58333 | 156521 | 23467 | 1345 | 1.58 |
| 1028 | Xinjiang | 6579942 | 28.00 | 2.56 | 201621 | 2686 | 84 | 32261 | 24343 | 129144 | 12124 | 6435 | 88699 | 94628 | 18420 | 2560 | 1.26 |
| 1029 | Tibet | 286242 | 31.81 | 2.45 | 8394 | 973 | 7 | 2930 | 4798 | 1580 | 47 | 12 | 5449 | 2227 | 1020 | 671 | 1.25 |
| 1030 | Yunnan | 6548268 | 31.27 | 2.59 | 200602 | 7122 | 172 | 21262 | 45555 | 93027 | 36704 | 11176 | 102015 | 85386 | 13317 | 7006 | 1.22 |
| 1031 | Zhejiang | 21735537 | 30.97 | 2.54 | 675858 | 19305 | 774 | 80859 | 193447 | 332899 | 50666 | 37292 | 220048 | 393843 | 74559 | 6713 | 1.23 |
| | | | | | | | | **township** | | | | | | | | | |
| 2001 | Anhui | 13378847 | 32.20 | 2.95 | 355306 | 19130 | 477 | 144219 | 160370 | 67744 | 1426 | 677 | 95625 | 182264 | 91921 | 4626 | 1.21 |
| 2002 | Beijing | 1548170 | 33.20 | 2.52 | 41959 | 1129 | 143 | 21808 | 2812 | 16414 | 710 | 1344 | 6224 | 20550 | 15964 | 350 | 1.42 |
| 2003 | Chongqing | 6401393 | 34.91 | 2.73 | 187287 | 7816 | 357 | 35957 | 71385 | 40448 | 41156 | 6157 | 46425 | 112018 | 23805 | 12855 | 1.20 |
| 2004 | Fujian | 8618108 | 37.67 | 3.09 | 224647 | 11851 | 318 | 44154 | 105240 | 65529 | 18822 | 2753 | 100650 | 83984 | 28551 | 23313 | 1.18 |
| 2005 | Gansu | 3941847 | 25.92 | 3.17 | 101071 | 5160 | 124 | 58128 | 13450 | 30226 | 4198 | 229 | 31721 | 30839 | 34944 | 8727 | 1.17 |



| Code | Region | | | | | | | rural | | | | | | | | | |
|---|---|---|---|---|---|---|---|---|---|---|---|---|---|---|---|---|---|
| 2006 | Guangdong | 17952939 | 26.41 | 3.52 | 357650 | 15136 | 348 | 119634 | 161452 | 60743 | 27235 | 3722 | 124661 | 175520 | 63890 | 8715 | 1.37 |
| 2007 | Guangxi | 10219075 | 34.43 | 3.34 | 264485 | 12263 | 480 | 94666 | 111560 | 58971 | 11002 | 549 | 53729 | 175149 | 42500 | 5370 | 1.10 |
| 2008 | Guizhou | 6164328 | 28.39 | 3.12 | 159970 | 12522 | 41 | 65929 | 60006 | 34332 | 11785 | 440 | 89287 | 28725 | 10464 | | 1.15 |
| 2009 | Hainan | 1988812 | 23.78 | 3.42 | 45035 | 2592 | 51 | 26889 | 15458 | 4359 | 607 | 314 | 19912 | 12356 | 14449 | 910 | 1.22 |
| 2010 | Hebei | 17725642 | 30.74 | 3.40 | 454034 | 12232 | 203 | 338450 | 45232 | 73026 | 3484 | 6074 | 90952 | 204531 | 5032 | | 1.12 |
| 2011 | Heilongjiang | 7328148 | 22.67 | 2.63 | 230438 | 7764 | 526 | 152211 | 54825 | 16851 | 604 | | 70838 | 130084 | 10411 | 5032 | 1.17 |
| 2012 | Henan | 18087162 | 32.04 | 3.60 | 435993 | 14307 | 304 | 242151 | 151413 | 53669 | 2676 | 391 | 91696 | 240373 | 114219 | 4012 | 1.11 |
| 2013 | Hubei | 10290017 | 38.10 | 3.12 | 267951 | 11284 | 318 | 136106 | 59020 | 18152 | 806 | | 75159 | 150951 | 47125 | 6000 | 1.18 |
| 2014 | Hunan | 15931187 | 36.74 | 3.18 | 413160 | 16084 | 1397 | 107304 | 216464 | 90305 | 12926 | 2245 | 103618 | 92116 | 8342 | | 1.16 |
| 2015 | Jiangsu | 17597864 | 39.53 | 3.00 | 493818 | 16021 | 436 | 194665 | 224247 | 86379 | 2299 | 2249 | 99148 | 142526 | 13226 | | 1.15 |
| 2016 | Jiangxi | 12543925 | 33.57 | 3.54 | 283781 | 10796 | 1125 | 57795 | 138466 | 80093 | 17102 | 1121 | 144491 | 98662 | 45425 | 5999 | 1.20 |
| 2017 | Jilin | 4484285 | 22.51 | 2.70 | 139477 | 4710 | 1966 | 90313 | 37025 | 6460 | 228 | | 34567 | 73754 | 5399 | | 1.14 |
| 2018 | Liaoning | 5200437 | 26.23 | 2.75 | 168663 | 5618 | 94 | 100064 | 11565 | 9229 | 1500 | | 51280 | 52098 | 1088 | | 1.08 |
| 2019 | Inner Mongolia | 5919165 | 24.38 | 2.74 | 172725 | 9637 | 1622 | 124351 | 14566 | 1422 | 191 | | 43195 | 35332 | 12852 | | 1.17 |
| 2020 | Ningxia | 1041959 | 24.82 | 3.14 | 25273 | 1397 | 58 | 16542 | 2590 | 176 | 54 | | 7109 | 12255 | 1166 | | 1.24 |
| 2021 | Qinghai | 1237394 | 21.94 | 3.06 | 28364 | 1806 | 1694 | 15491 | 7308 | 386 | 30 | | 6140 | 9814 | 2946 | | 1.27 |
| 2022 | Shaanxi | 8394596 | 28.85 | 3.05 | 218969 | 10349 | 295 | 103810 | 63776 | 53427 | 6133 | 2172 | 61288 | 115983 | 30075 | 21972 | 1.20 |
| 2023 | Shandong | 19633371 | 32.14 | 3.03 | 555539 | 16773 | 117 | 412345 | 102936 | 86379 | 2235 | 935 | 105549 | 177664 | 274908 | 14191 | 1.13 |
| 2024 | Shanghai | 3391859 | 30.25 | 2.45 | 100049 | 3066 | 715 | 57795 | 44272 | 29262 | 638 | 4710 | 35992 | 46750 | 19423 | 950 | 1.33 |
| 2025 | Shanxi | 8098814 | 25.43 | 3.24 | 208837 | 7124 | 292 | 128133 | 41454 | 42626 | 2929 | 819 | 49930 | 87194 | 12419 | 5999 | 1.16 |
| 2026 | Sichuan | 16241360 | 34.47 | 2.80 | 494678 | 24545 | 2048 | 133695 | 170345 | 141458 | 9146 | 64579 | 144800 | 259633 | 80423 | 34367 | 1.11 |
| 2027 | Tianjin | 1605727 | 29.64 | 2.98 | 36626 | 688 | 6 | 20978 | 1965 | 559 | 1085 | | 5896 | 13066 | 1817 | 135 | 1.44 |
| 2028 | Xinjiang | 3556387 | 26.04 | 2.75 | 95090 | 2368 | 50 | 57285 | 7087 | 301 | 187 | | 31109 | 21827 | 34576 | 9946 | 1.32 |
| 2029 | Tibet | 444301 | 33.52 | 2.89 | 10835 | 1334 | 69 | 5712 | 5333 | 1058 | 39 | 27 | 5633 | 2406 | 2961 | 1169 | 1.26 |
| 2030 | Yunnan | 9949242 | 30.04 | 3.29 | 249892 | 15089 | 538 | 95990 | 113777 | 49076 | 5598 | 540 | 85728 | 73181 | 58444 | 47628 | 1.14 |
| 2031 | Zhejiang | 14035213 | 38.53 | 2.66 | 435571 | 17019 | 321 | 78393 | 215994 | 143891 | 9590 | 4722 | 88524 | 262572 | 92204 | 9290 | 1.16 |
| 3001 | Anhui | 33860554 | 34.04 | 3.12 | 972114 | 12697 | 1032 | 594442 | 384935 | 5062 | 259 | 113 | 122416 | 440296 | 399437 | 22662 | 1.10 |
| 3002 | Beijing | 3289036 | 35.39 | 2.76 | 85494 | 2139 | 89 | 81788 | 2698 | 2877 | 93 | 177 | 19546 | 63298 | 1798 | | 1.36 |
| 3003 | Chongqing | 13078118 | 42.04 | 2.72 | 436237 | 8496 | 810 | 215548 | 219389 | 6337 | 3076 | 383 | 34275 | 160849 | 102717 | 2991 | 1.08 |
| 3004 | Fujian | 16018762 | 41.24 | 3.16 | 447940 | 13851 | 615 | 152099 | 279696 | 27946 | 1860 | 190 | 105558 | 152003 | 108638 | 95592 | 1.11 |
| 3005 | Gansu | 16451585 | 21.94 | 3.89 | 444734 | 2789 | 233 | 434394 | 12043 | 911 | 94 | 81 | 23583 | 50990 | 139709 | | 0.94 |
| 3006 | Guangdong | 38064798 | 25.99 | 3.74 | 825588 | 7932 | 862 | 473821 | 328499 | 27016 | 3542 | 642 | 168179 | 388958 | 244088 | 32295 | 1.22 |
| 3007 | Guangxi | 28011829 | 28.82 | 3.47 | 784492 | 7837 | 834 | 494076 | 294396 | 7474 | 300 | 83 | 100152 | 424423 | 210891 | 60843 | 1.01 |
| 3008 | Guizhou | 22784212 | 27.92 | 3.29 | 657275 | 13176 | 244 | 526145 | 137494 | 5485 | 1206 | 121 | 80232 | 408026 | 247780 | 134413 | 1.03 |
| 3009 | Hainan | 4359920 | 21.29 | 3.63 | 109378 | 771 | 69 | 101212 | 8248 | 437 | 217 | 35 | 16584 | 68949 | 2307 | | 1.09 |
| 3010 | Hebei | 41530827 | 30.09 | 3.50 | 1138877 | 6755 | 525 | 1108487 | 32754 | 3591 | 510 | 290 | 65563 | 351042 | 689663 | 39364 | 1.04 |
| 3011 | Heilongjiang | 17281672 | 20.92 | 3.19 | 472849 | 3926 | 1647 | 469755 | 3174 | 2668 | 1148 | 30 | 5933 | 44163 | 399849 | 86830 | 1.13 |
| 3012 | Henan | 58410084 | 32.23 | 3.58 | 1593259 | 18790 | 715 | 1263614 | 341472 | 6231 | 554 | 178 | 170146 | 632719 | 632719 | 30697 | 1.01 |
| 3013 | Hubei | 28154883 | 38.64 | 3.40 | 805308 | 11381 | 807 | 395220 | 405959 | 12191 | 2267 | 1052 | 87280 | 373421 | 286599 | 69389 | 1.01 |
| 3014 | Hunan | 37743917 | 34.27 | 3.54 | 1008324 | 9900 | 2170 | 496152 | 516168 | 5569 | 262 | 73 | 113888 | 408562 | 427367 | 68407 | 1.04 |
| 3015 | Jiangsu | 31993485 | 42.35 | 3.03 | 978352 | 13096 | 999 | 526012 | 444382 | 17344 | 893 | 2817 | 77218 | 494838 | 411206 | 8186 | 1.06 |
| 3016 | Jiangxi | 26200474 | 33.81 | 3.86 | 627420 | 6578 | 1410 | 251425 | 373710 | 8390 | 355 | 118 | 184327 | 209487 | 198186 | 41998 | 1.07 |
| 3017 | Jilin | 12896125 | 20.98 | 3.35 | 353543 | 2220 | 2523 | 347297 | 3170 | 4561 | 676 | 59 | 11283 | 35524 | 274007 | 34949 | 1.07 |
| 3018 | Liaoning | 16667944 | 25.95 | 3.12 | 519784 | 3994 | 237 | 512930 | 6643 | 3709 | 390 | 106 | 31856 | 123657 | 360371 | 7894 | 1.02 |
| 3019 | Inner Mongolia | 11371410 | 22.17 | 2.97 | 337168 | 4773 | 1167 | 331674 | 6301 | 3644 | 77 | 245 | 10616 | 34647 | 206674 | 9004 | 1.12 |




| Prov_id | Province |  |  |  |  |  |  |  |  |  |  |  |  |  |  |  | Amp. factor |
|---|---|---|---|---|---|---|---|---|---|---|---|---|---|---|---|---|---|
| 3020 | Ningxia | 3514019 | 22.12 | 3.54 | 86461 | 1371 | 35 | 80927 | 1965 | 64 | 13 | 4863 | 4944 | 9056 | 60381 | 13451 | 1.13 |
| 3021 | Qinghai | 3331549 | 18.51 | 4.06 | 71842 | 604 | 1521 | 69459 | 2789 | 181 | 7 | 10 | 2675 | 9718 | 36221 | 23832 | 1.11 |
| 3022 | Shaanxi | 20681076 | 31.22 | 3.54 | 572916 | 6711 | 497 | 481090 | 94599 | 3360 | 348 | 230 | 60338 | 235474 | 142395 | 141420 | 1.01 |
| 3023 | Shandong | 49111245 | 31.95 | 3.07 | 1549890 | 8748 | 182 | 1511164 | 40165 | 6807 | 399 | 103 | 77610 | 400711 | 1025247 | 55070 | 1.03 |
| 3024 | Shanghai | 2868506 | 38.83 | 2.37 | 90972 | 1752 | 1153 | 31644 | 57352 | 3415 | 49 | 264 | 8884 | 48551 | 33963 | 1326 | 1.29 |
| 3025 | Shanxi | 19383034 | 25.09 | 3.44 | 521669 | 4921 | 593 | 481296 | 38553 | 6348 | 290 | 103 | 34053 | 138101 | 243316 | 111120 | 1.07 |
| 3026 | Sichuan | 47509769 | 36.63 | 3.10 | 1625052 | 36122 | 3253 | 1067677 | 574735 | 16573 | 1425 | 764 | 147168 | 513785 | 611594 | 388627 | 0.92 |
| 3027 | Tianjin | 3005963 | 25.95 | 3.21 | 78318 | 570 | 30 | 74498 | 686 | 3345 | 110 | 249 | 2325 | 7772 | 68306 | 485 | 1.19 |
| 3028 | Xinjiang | 13519120 | 22.35 | 3.55 | 314397 | 2226 | 115 | 309505 | 2663 | 4345 | 82 | 28 | 1730 | 36704 | 207565 | 60624 | 1.20 |
| 3029 | Tibet | 2461371 | 27.55 | 4.95 | 44816 | 1260 | 718 | 27819 | 17858 | 360 | 26 | 13 | 2594 | 23631 | 5152 | 14699 | 1.06 |
| 3030 | Yunnan | 30970894 | 25.61 | 3.89 | 756974 | 10742 | 1276 | 461191 | 296513 | 6950 | 2470 | 592 | 68863 | 239753 | 112129 | 346971 | 1.04 |
| 3031 | Zhejiang | 22249067 | 49.12 | 2.67 | 740469 | 17587 | 807 | 152558 | 544733 | 58732 | 1649 | 384 | 60829 | 419761 | 236627 | 40839 | 1.10 |

Note: The three **urbanity attributes**, namely **urban/township/rural**, are represented by number 1/2/3 in the first column of this table; "**Prov_id**" refers to the ID number of each province; "**Aver. pop. per family**" refers to the average number of population per family; "**Amp. factor**" refers to the amplification factor used to amplify the building related statistics from 2010 to 2015 (see **Sect. 2.1** and **2.4.1** for more details).




**Table 3**: The population proportions and thresholds used for each province to assign the grids in the 2015 GHSL profile with urban/township/rural attributes.

| Province | Province ID | 2010-census recorded population in each urbanity | | | | Population proportion | | | Population threshold (PT) | |
| --- | --- | --- | --- | --- | --- | --- | --- | --- | --- | --- |
| | | urban | township | rural | sum | urban | township | rural | PT1 (urban/township) | PT2 (township/rural) |
| Anhui | 01 | 12182587 | 13394530 | 33923351 | 59500468 | 20.47% | 22.51% | 57.01% | 13950 | 6907 |
| Beijing | 02 | 15563215 | 1295477 | 2753676 | 19612368 | 79.35% | 6.61% | 14.04% | 2702 | 1775 |
| Chongqing | 03 | 8681611 | 6614192 | 13550367 | 28846170 | 30.10% | 22.93% | 46.97% | 11194 | 5412 |
| Fujian | 04 | 12548384 | 8513556 | 15832277 | 36894217 | 34.01% | 23.08% | 42.91% | 6020 | 2586 |
| Gansu | 05 | 5259935 | 3932250 | 16384078 | 25575263 | 20.56% | 15.38% | 64.06% | 15167 | 9337 |
| Guangdong | 06 | 52388382 | 16641873 | 35290204 | 104320459 | 50.22% | 15.95% | 33.83% | 5229 | 2996 |
| Guangxi | 07 | 8352777 | 10065066 | 27605918 | 46023761 | 18.15% | 21.87% | 59.98% | 11694 | 5065 |
| Guizhou | 08 | 5537562 | 6199971 | 23011023 | 34748556 | 15.94% | 17.84% | 66.22% | 18152 | 10413 |
| Hainan | 09 | 2324288 | 1984228 | 4362969 | 8671485 | 26.80% | 22.88% | 50.31% | 8256 | 3679 |
| Hebei | 10 | 14388021 | 17187307 | 40278882 | 71854210 | 20.02% | 23.92% | 56.06% | 5682 | 2403 |
| Heilongjiang | 11 | 14122516 | 7201199 | 16990276 | 38313991 | 36.86% | 18.80% | 44.34% | 3848 | 1485 |
| Henan | 12 | 18331493 | 17888274 | 57810172 | 94029939 | 19.50% | 19.02% | 61.48% | 15199 | 8456 |
| Hubei | 13 | 17928160 | 10516925 | 28792642 | 57237727 | 31.32% | 18.37% | 50.30% | 11667 | 6345 |
| Hunan | 14 | 12738442 | 15714621 | 37247699 | 65700762 | 19.39% | 23.92% | 56.69% | 13552 | 5876 |
| Jiangsu | 15 | 30166466 | 17205022 | 31289453 | 78660941 | 38.35% | 21.87% | 39.78% | 6559 | 3341 |
| Jiangxi | 16 | 7500291 | 11995669 | 25067837 | 44567797 | 16.84% | 26.92% | 56.25% | 11326 | 3400 |
| Jilin | 17 | 10196745 | 4451454 | 12804616 | 27452815 | 37.14% | 16.21% | 46.64% | 6168 | 2866 |
| Liaoning | 18 | 22021184 | 5166779 | 16558360 | 43746323 | 50.34% | 11.81% | 37.85% | 3511 | 1882 |
| Inner Mongolia | 19 | 8011564 | 5708610 | 10986117 | 24706291 | 32.43% | 23.11% | 44.47% | 11152 | 5036 |
| Ningxia | 20 | 2059295 | 962727 | 3279228 | 6301350 | 32.68% | 15.28% | 52.04% | 11659 | 7624 |
| Qinghai | 21 | 1368033 | 1148221 | 3110469 | 5626723 | 24.31% | 20.41% | 55.28% | 11850 | 5113 |
| Shaanxi | 22 | 8837175 | 8222162 | 20268042 | 37327379 | 23.67% | 22.03% | 54.30% | 13731 | 6872 |
| Shandong | 23 | 28364984 | 19255743 | 48171992 | 95792719 | 29.61% | 20.10% | 50.29% | 6577 | 3372 |
| Shanghai | 24 | 17640842 | 2914256 | 2464098 | 23019196 | 76.64% | 12.66% | 10.70% | 4936 | 2750 |
| Shanxi | 25 | 9414053 | 7746486 | 18551562 | 35712101 | 26.36% | 21.69% | 51.95% | 8804 | 3890 |
| Sichuan | 26 | 15915660 | 16428768 | 48073100 | 80417528 | 19.79% | 20.43% | 59.78% | 14668 | 8123 |
| Tianjin | 27 | 8858126 | 1419767 | 2660800 | 12938693 | 68.46% | 10.97% | 20.56% | 3138 | 1872 |
| Xinjiang | 28 | 6071803 | 3263949 | 12480063 | 21815815 | 27.83% | 14.96% | 57.21% | 10473 | 3620 |
| Tibet | 29 | 272322 | 408267 | 2321576 | 3002165 | 9.07% | 13.60% | 77.33% | 9751 | 4522 |
| Yunnan | 30 | 6324830 | 9634242 | 30007694 | 45966766 | 13.76% | 20.96% | 65.28% | 19028 | 8699 |
| Zhejiang | 31 | 20386294 | 13163915 | 20876682 | 54426891 | 37.46% | 24.19% | 38.36% | 5599 | 2513 |

Note: For each province, **"PT1(urban/township)"** and **"PT2 (township/rural)"** are the population thresholds to assign the grids in the 2015 GHSL profile with urban/township/rural attributes. According to the population density $\lambda$ in each grid, the assignment criteria are that: if $\lambda \geq PT1$, the grid is assigned as **urban**; if $PT1 > \lambda \geq PT2$, **township**; if $\lambda < PT2$, **rural** (see context in **Sect. 2.3** for more details).



**Table 4**: Average unit construction price (per m$^2$) for each of the 17 building subtypes used in this paper.

| Structure type | Storey class | Building subtype abbreviation | Unit construction price (RMB/m$^2$ in 2015 current price) |
|---|---|---|---|
| brick/wood | 1 | BRIWOMC1 | 2050 |
| | 2-3 | BRIWOMC23 | 2350 |
| steel/RC | 1 | STLRCMC1 | 3700 |
| | 2-3 | STLRCMC23 | 3900 |
| | 4-6 | STLRCMC46 | 4100 |
| | 7-9 | STLRCMC79 | 4300 |
| | ≥10 | STLRCMC10 | 4500 |
| mixed | 1 | MIXEDMC1 | 2800 |
| | 2-3 | MIXEDMC23 | 3000 |
| | 4-6 | MIXEDMC46 | 3200 |
| | 7-9 | MIXEDMC79 | 3400 |
| | ≥10 | MIXEDMC10 | 3600 |
| others | 1 | OTHERMC1 | 2600 |
| | 2-3 | OTHERMC23 | 2800 |
| | 4-6 | OTHERMC46 | 3000 |
| | 7-9 | OTHERMC79 | 3200 |
| | ≥10 | OTHERMC10 | 3400 |





**Table 5**: The modeled floor area and replacement value of residential buildings in urban/township/rural urbanity of the 31 provinces in mainland China.

| Province ID | Province name | Modeled residential building floor area (**million m²**) in each urbanity level | | | Modeled residential building replacement value (**billion RMB**, in 2015 current price) in each urbanity level | | |
|---|---|---|---|---|---|---|---|
| | | urban | township | rural | urban | township | rural |
| 01 | Anhui | 357 | 431 | 1150 | 507 | 498 | 1080 |
| 02 | Beijing | 516 | 51 | 117 | 1920 | 147 | 223 |
| 03 | Chongqing | 250 | 222 | 550 | 564 | 428 | 825 |
| 04 | Fujian | 377 | 326 | 667 | 1000 | 648 | 1240 |
| 05 | Gansu | 141 | 102 | 351 | 231 | 114 | 259 |
| 06 | Guangdong | 1640 | 448 | 864 | 4130 | 798 | 1060 |
| 07 | Guangxi | 260 | 350 | 808 | 618 | 691 | 1160 |
| 08 | Guizhou | 143 | 175 | 635 | 221 | 197 | 487 |
| 09 | Hainan | 60 | 47 | 86 | 141 | 79 | 89 |
| 10 | Hebei | 448 | 544 | 1210 | 916 | 880 | 1370 |
| 11 | Heilongjiang | 341 | 166 | 360 | 844 | 257 | 365 |
| 12 | Henan | 630 | 580 | 1880 | 1120 | 1020 | 2550 |
| 13 | Hubei | 582 | 392 | 1090 | 1270 | 610 | 1400 |
| 14 | Hunan | 431 | 583 | 1290 | 749 | 786 | 1360 |
| 15 | Jiangsu | 1040 | 695 | 1350 | 3250 | 1910 | 3130 |
| 16 | Jiangxi | 234 | 419 | 884 | 387 | 533 | 845 |
| 17 | Jilin | 258 | 100 | 266 | 1080 | 268 | 483 |
| 18 | Liaoning | 572 | 136 | 426 | 2080 | 353 | 710 |
| 19 | Inner Mongolia | 206 | 143 | 247 | 1170 | 485 | 559 |
| 20 | Ningxia | 63 | 26 | 78 | 185 | 56 | 121 |
| 21 | Qinghai | 41 | 26 | 60 | 107 | 55 | 87 |
| 22 | Shaanxi | 260 | 242 | 644 | 597 | 523 | 960 |
| 23 | Shandong | 936 | 632 | 1530 | 2450 | 1380 | 2480 |
| 24 | Shanghai | 516 | 102 | 109 | 2120 | 339 | 254 |
| 25 | Shanxi | 255 | 206 | 484 | 661 | 361 | 587 |
| 26 | Sichuan | 483 | 556 | 1740 | 795 | 780 | 1780 |
| 27 | Tianjin | 255 | 48 | 78 | 1000 | 204 | 217 |
| 28 | Xinjiang | 184 | 92 | 299 | 516 | 206 | 279 |
| 29 | Tibet | 9 | 15 | 67 | 25 | 35 | 83 |
| 30 | Yunnan | 221 | 312 | 767 | 334 | 431 | 727 |
| 31 | Zhejiang | 673 | 542 | 1090 | 1820 | 1200 | 1910 |
| In total: | | 12400 | 8710 | 21200 | 32808 | 16300 | 28700 |

Note: (a) In this paper, for each of the 17 building subtypes in each grid, the same unit construction price is used to derive the replacement value in different urbanities and provinces; (b) The modeled floor area and replacement value are for residential buildings (see context in **Sect. 3.1.1** for more details).



**Table 6**: The regression parameters and correlation coefficients for population and floor area in each province.

| Province ID | Province name | Pop_a | Pop_b | Pop_R² | FloorArea_a | FloorArea_b | Area_R² |
|---|---|---|---|---|---|---|---|
| 01 | Anhui | 1.227 | -121096 | 0.9525 | 1.2256 | -4000000 | 0.917 |
| 02 | Beijing | 1.4375 | -11276 | 0.9986 | 1.4947 | -3000000 | 0.9993 |
| 03 | Chongqing | 1.1261 | -68344 | 0.9624 | 1.2336 | -6000000 | 0.9049 |
| 04 | Fujian | 1.2485 | -66004 | 0.9741 | 0.9975 | 2000000 | 0.8165 |
| 05 | Gansu | 1.1977 | -38495 | 0.9876 | 1.1499 | -651568 | 0.9526 |
| 06 | Guangdong | 1.5014 | -212584 | 0.9712 | 1.6419 | -9000000 | 0.9285 |
| 07 | Guangxi | 0.936 | 43874 | 0.9251 | 0.9482 | 993643 | 0.8633 |
| 08 | Guizhou | 1.1151 | -37198 | 0.99 | 1.2213 | -2000000 | 0.961 |
| 09 | Hainan | 1.2608 | -80398 | 0.9692 | 1.2068 | -2000000 | 0.9675 |
| 10 | Hebei | 1.1402 | -27316 | 0.9832 | 1.05 | 184103 | 0.9276 |
| 11 | Heilongjiang | 1.1307 | -30556 | 0.9839 | 1.0486 | 118704 | 0.977 |
| 12 | Henan | 1.1817 | -93834 | 0.9599 | 1.0788 | -554637 | 0.9039 |
| 13 | Hubei | 1.2252 | -101914 | 0.9788 | 1.374 | -7000000 | 0.9387 |
| 14 | Hunan | 1.1237 | -212458 | 0.9628 | 1.032 | 6000000 | 0.8858 |
| 15 | Jiangsu | 1.3726 | -266170 | 0.9335 | 1.2612 | 6000000 | 0.7783 |
| 16 | Jiangxi | 1.1411 | -18384 | 0.9901 | 1.0855 | 252638 | 0.9365 |
| 17 | Jilin | 1.0739 | -16159 | 0.9907 | 0.9804 | 715875 | 0.9894 |
| 18 | Liaoning | 1.1467 | -273787 | 0.9957 | 1.0608 | -933912 | 0.9902 |
| 19 | Inner Mongolia | 1.1574 | -11718 | 0.9814 | 1.1262 | -162051 | 0.978 |
| 20 | Ningxia Hui | 1.2559 | -37867 | 0.9668 | 1.0727 | 507343 | 0.9588 |
| 21 | Qinghai | 1.1457 | -1152.1 | 0.9935 | 0.9763 | 377230 | 0.9851 |
| 22 | Shaanxi | 1.2448 | -53315 | 0.9857 | 1.2304 | -1000000 | 0.9459 |
| 23 | Shandong | 1.1272 | -35525 | 0.9725 | 1.0518 | 392271 | 0.934 |
| 24 | Shanghai | 1.1752 | 286962 | 0.9665 | 1.2034 | 6000000 | 0.9368 |
| 25 | Shanxi | 1.2375 | -38478 | 0.9904 | 1.1738 | -474998 | 0.9456 |
| 26 | Sichuan | 1.175 | -478703 | 0.9754 | 1.0902 | -7000000 | 0.9561 |
| 27 | Tianjin | 1.1832 | 274914 | 0.8724 | 1.2782 | 4000000 | 0.8993 |
| 28 | Xinjiang | 1.1519 | -2241.9 | 0.9827 | 1.1454 | -10818 | 0.9789 |
| 29 | Tibet | 1.2168 | -3498.3 | 0.9834 | 1.1196 | -1699.8 | 0.9199 |
| 30 | Yunnan | 1.1632 | -26658 | 0.9898 | 0.9589 | 1000000 | 0.9083 |
| 31 | Zhejiang | 1.2686 | -45842 | 0.9751 | 1.323 | -4000000 | 0.88 |

Note: "**Pop_a**" and "**Pop_b**" are the linear regression parameters between the 2015 GHSL population and the 2010-census recorded population; "**FloorArea_a**" and "**FloorArea_b**" are the linear regression parameters between the modeled residential building floor area in this paper and that extracted from the 2010-census records; "**Pop_R²**" and "**FloorArea_R²**" are the correlation coefficients of population and floor area, respectively. For Hunan, Liaoning, and Sichuan provinces, the population and floor area comparisons are compared at the prefecture-level; while for the other 28 provinces, the population and floor area comparisons are at the county-level. The correlation analysis figures for each of the 31 provinces are available from the online supplement (see the context in **Sect. 3.2.2** for more details).


## Appendix A

For each grid, to derive the population living in each of the 17 building subtypes (their abbreviations are given in
Table 4), namely the 17 to-be-solved variables on the left side of the equation set in Sect. 2.4.2., a series of
distribution steps based on a prioritized ranking of building types and storey classes are used in this paper. A
MATLAB script and an input file illustrating the distribution processes are also available from the online
supplement. With the help of the MATLAB script, it will be easier to understand the distribution steps as follows.

(1) For brick/wood structure type, in each grid if $Num_{BRIWO} < Num_{storey1}$, the population living in brick/wood
structure types ($Num_{BRIWO}$) is first placed into the 1-storey class, then we get $BRIWOMC1 = Num_{BRIWO}$
and the remaining population living in brick/wood structure type is 0, while the remaining population living
in the 1-storey class is ($Num_{storey1} - Num_{BRIWO}$); but if $Num_{BRIWO} \geq Num_{storey1}$, then the population
living in the 1 storey class buildings ($Num_{storey1}$) are assumed to be in brick/wood structure type, we get
$BRIWOMC1 = Num_{storey1}$ and the remaining population living in brick/wood buildings is ($Num_{BRIWO} -$
$Num_{storey1}$), while the remaining population living in the 1-storey class is 0;

(2) If the remaining population living in brick/wood buildings ($Num_{BRIWO} - Num_{storey1}$) $< Num_{storey23}$,
then they are placed into 2-3 storey class and we get $BRIWOMC23 = Num_{BRIWO} - BRIWOMC1$ or
$BRIWOMC23 = Num_{BRIWO} - Num_{storey1}$, and the remaining population in the 2-3 storey class is
($Num_{storey23} - (Num_{BRIWO} - Num_{storey1})$); but if ($Num_{BRIWO} - Num_{storey1}$) $\geq Num_{storey23}$, we
directly assign $BRIWOMC23 = Num_{storey23}$ and the remaining population living in brick/wood buildings
is ($Num_{BRIWO} - Num_{storey1} - Num_{storey23}$);

(3) For steel/RC structure type, in each grid if $Num_{STLRC} < Num_{storey\geq10}$, the population living in steel/RC
structure type ($Num_{STLRC}$) is first placed in the $\geq 10$ storey class, and we get $STLRCMC10 = Num_{STLRC}$,
then the remaining population living in the $\geq 10$ storey class is ($Num_{storey\geq10} - Num_{STLRC}$), while the
remaining population living in steel/RC structure type is 0; but if $Num_{STLRC} \geq Num_{storey\geq10}$, then we
directly assign $STLRCMC10 = Num_{storey\geq10}$, and the remaining population living in steel/RC structure type
is ($Num_{STLRC} - Num_{storey\geq10}$), while the remaining population living in $\geq 10$ storey class is 0;

(4) Following the above step (3), if $Num_{STLRC} \geq Num_{storey\geq10}$, the remaining population living in steel/RC
structure type is compared with the population living in other storey class and distributed into the remaining
storey classes from the highest to the lowest, assuming that the least population in steel/RC would be in the
1-storey class, then we get $STLRCMC79 = Num_{STLRC} - Num_{storey\geq10}$ or $STLRCMC79 = Num_{storey79}$
or $STLRCMC79 = 0$; $STLRCMC46 = Num_{STLRC} - Num_{storey\geq10} - Num_{storey79}$ or $STLRCMC46 =$
$Num_{storey46}$ or $STLRCMC46 = 0$; $STLRCMC23 = Num_{STLRC} - Num_{storey\geq10} - Num_{storey79} -$
$Num_{storey46}$ or $STLRCMC23 = Num_{storey23} - (Num_{BRIWO} - Num_{storey1})$ or $STLRCMC23 = 0$;
$STLRCMC1 = Num_{STLRC} - Num_{storey\geq10} - Num_{storey79} - Num_{storey46} - (Num_{storey23} -$
$(Num_{BRIWO} - Num_{storey1}))$ or $STLRCMC1 = (Num_{storey1} - Num_{BRIWO})$ or $STLRCMC1 = 0$;



(5)   After getting the population living in 7 building subtypes ($BRIWOMC1$, $BRIWOMC23$, $STLRCMC10$, $STLRCMC79, STLRCMC46, STLRCMC23, STLRCMC1$) and the remaining population living in each of the five storey classes determined, to derive the population living in storey class with structure type "mixed" and "other", we assume that the populations living in the five storey classes of "mixed" structure type are equal to the product of the remaining population in each storey class and the ratio of $Num_{MIXED}/(Num_{MIXED} + Num_{OTHER})$; similarly, the populations living in the five storey classes of "other" structure type are equal to the product of the remaining population in each storey class and the ratio of $Num_{OTHER}/(Num_{MIXED} + Num_{OTHER})$.