# Peer review of "Residential building stock modeling for mainland China targeted for seismic risk assessment"

_Natural Hazards and Earth System Sciences, 2021_

## Author Comment (AC1)

Comment on "Residential building stock modeling for mainland China targeted for seismic risk assessment" by Xin et al.

**Comment 1:** The authors present an interesting approach to achieve a nation-wide model for the building stock to be used in seismic risk assessment. Based on various statistical data and information derived from secondary sources and remotely-sensed data, they present a method to derive at a geo-coded 1km×1km resolution residential building exposure model for 31 provinces of mainland China. Moreover, based on a sensitivity analysis for one case study, the authors present possible sources of uncertainty in the results, and show how these may be decreased during future research efforts.

**Response:** Thank you very much for your time and efforts in reviewing this manuscript.

**Comment 2:** Overall, the paper is timely, and well-structured. The overall point of criticism is that the sole use of statistical data to derive at a "real world" building stock model neglects region-specific and very local impacts on the distribution and quality of assets, which may lead to systematic over- and underestimation in certain areas of the country.

**Response:** It is true that regional specific and very local impacts are not considered in this modeling process. This is mainly limited by the difficulty in collecting such detailed data for specific regions, which are usually proprietary and not publicly accessible.

**Comment 3:** Nevertheless, I strongly believe that the method is worth being published so that the international research community can further refine the method and decrease inherent uncertainties.

**Response:** Thanks for this affirmation. Our detailed responses to your comments are as follows.

Some minor comments:

**Comment 4:** Line 152/153: something is missing here. Should be re-formulated.

**Response:** Thanks for pointing this out. The complete reason that the township/street level population generated by using the multi-variate regression method in Fu et al. (2014a) tends to overpredict the population density in a

sparsely populated area and underpredict the population density in a densely populated area is enriched as follows:

"The reasons for such discrepancies are that: (1) The population density developed for each land use type by using the multi-variate regression method is the average population density, thus the over/under prediction of the actual population density in certain areas is inevitable; (2) When applying the multi-variate regression method, no additional supplementary data (e.g., road density, nighttime light) is employed to adjust the development level difference in different regions, because the development level is much higher than the average in places such as the downtown area of metropolitan cities like Shenzhen and Guangzhou."

We will add the above explanation to the revised manuscript.

**Comment 5:** Line 193/194: could be better formulated.

**Response:** Thanks for pointing this out. The old expression "*The census for the year 2020 is just initiated and normally it takes around two years to publish the final surveyed data. Therefore, the current latest census data are for the year 2010*" will be rephrased as "*Detailed statistics for the year 2020 are not publicly accessible yet. Therefore, census data for the year 2010 will be used to elaborate the modeling process*".

**Comment 6:** Line 270/271: please elaborate a bit more why the spatial coverage is limited.

**Response:** Thanks for pointing this out. The limited spatial coverage of PopGrid China is related to its development method, namely the multi-variate regression method (Fu et al., 2014a). In this method, it was assumed that the spatial distribution of population is limited within the six land use types recognized from the Landsat TM images, namely cultivated land, forest land, grass land, rural residential land, urban residential land, industrial and transportation land. However, in actual cases, population distribute more widely and are beyond these land use types.

**Comment 7:** Line 469: in times of almost unlimited computing capacity, this should not be an issue. In contrast, applying the same unit price over the entire area of (mainland) China is a major source of uncertainty of the method, which should be addressed in more detail in the respective section 4.

**Response:** Thanks for pointing this out. In Line 467-469, we emphasized that "*There are significant differences across the country in terms of economic development level, geographic climatic diversity, and standardization in building construction. Therefore, it is mainly for computational convenience that this paper applies the same unit construction price for all the provinces and all the urbanity levels.*". The "computation convenience" here does not refer to the limitation in computing capacity of hardware. Instead, we mean it is convenient to only use a uniform price list to preliminarily calculate the replacement value of residential buildings in each grid, since it is quite difficult to compile a complete and accurate building construction price list for different provinces and urbanities. On one hand, this is because different documents include different items in calculating their unit construction prices. For example, some only consider the cost to build the main structure, while others also consider the cost of supporting facility and landscaping. On the other hand, different areas have different seismic fortification requirement, which will also alter the unit construction price for the same type of building.

Actually, before compiling the unit construction price in Table 4 of the manuscript, we have consulted cost-engineers from the real-estate industry. Their internal documents on cost control indicate that for the same type of residential building, the unit construction price of the main structure in different regions in China is limited to 300 RMB. Also, according to Li et al. (2021, Earthquake Spectra), the average unit construction price of multi-story reinforced concrete in urban areas of Tibet reaches 3200 RMB/m$^2$, which is quite comparable to the unit construction price for the same type of building in those more developed coastal areas.

Therefore, we prefer to only provide a reference set of unit construction price in Table 4 and avoid to over-manipulate it. As you point out, this will cause major uncertainty and we totally agree. However, for the sake of model users, this simplicity and transparency will make it more convenient for them to adjust the reference price to their targeted study area by simply multiplying some rectification factors.

We will add the above explanation to the limitation discussion in Section 4.

**Comment 8:** Section 3.1.2: Here it is not clear to me what the key message is; obviously, higher buildings will have a higher density of floor area and, thus, a higher population density. My suggestion is to elaborate this a bit more, or to delete this particular section.

**Response:** Thanks for this comment. As implied from the title of this subsection, this short section serves as an example to demonstrate the modeled floor area in Shanghai, which is to help potential readers to conduct direct comparison with other reports or modeling results. For example, they can directly check whether the grids that have the highest modeled floor area are within those most prosperous regions in Shanghai, as explained in Line 474-476 of the manuscript.

**Comment 9:** Figure 1: technically, the classes are not clearly distinguishable (what if a grid has exactly 4936 or 2750 inhabitants?), please adjust.

**Response:** Thanks for pointing this out. We think you refer to Figure 2. The modified population ranges for urban/township/rural urbanity levels in Figure 2 are as follows.

[Figure]

**Comment 10:** Figure 4 (and related section in the main text body): from my understanding it would be more explanatory how well your method is suitable for application if you would compute the differences between the modelled floor area per km^2 and the 3d view provided in inlet (c), also in terms of uncertainty

quantification. Please also consider the similar issue of classes given already for Figure 1 (and check all the other Figures, also in Figure 9 this is wrong).

**Response:** We are afraid there is some misunderstanding here. Indeed, the population data in Figure 2 (not Figure 1) and Figure 4(c) are from the 2015 GHSL developed by the European Union. And this population density profile is the base for us to divide the grids in each province into urban/township/rural levels. After this, we further disaggregate 2010-census statistics for urban/township/rural levels into corresponding grids.

Therefore, Figure 2 is to demonstrate how we assign urban/township/rural attributes to grids according to 2015 GHSL population density. Figure 4(a) is the example demonstrating one of our modeled products, namely the floor area in each grid. Figure 4(c) is plotted on the base of Figure 4(a), and its height is determined by the 2015 GHSL population in these grids, but the floor area is the same as that in Figure 4(a). By plotting Figure 4(c), we mainly want to show the location of those most densely populated grids.

Figure 9 is quite different from Figure 2 or Figure 4, because it is an application of the modeled results. It gives the seismic loss ratios calculated based our modeled building floor area, replacement value as well as an empirical vulnerability curve and the intensity map of the 2008 Ms8.0 Wenchuan earthquake.

**Comment 11:** Given these constraints I recommend revisions before the manuscript may become acceptable for publication.

**Response:** We deeply appreciate your generosity in spending time on reviewing this manuscript, which requires a lot of patience and efforts. We hope our responses have solved your concerns on this work. If not, we would like to make further explanation.

---

## Author Comment (AC4)

This paper basically proposed a downscaling approach to allocating building stock per province in China for better risk assessment, which well fits the scope of this journal. Given current revision status of this paper, I only have one main concern, which is about the definition of urban/township/rural. In reality, we can hardly differentiate them, in particular at pixelated level. Shanghai as we all know is highly urbanized, but still I can see many rural pixels from Figure 1, I do not believe it is the real case here. To me, 'rural' is mostly remote natural areas, I assume you intend to say 'village'.

**Response:** Thanks for your review. We totally understand your concern. It is true that in those highly urbanized provinces (e.g., Shanghai, Beijing, Guangzhou, Zhejiang, Jiangsu etc.), rural areas are not remote natural areas, although this is true for mountainous provinces (e.g., Sichuan, Guizhou, Yunnan, Tibet etc.). However, we consider it is also not appropriate to change "rural" to "village" in the context of this article, since villages typically have administrative boundaries. As explained in Section 2.3 of the manuscript, "The urbanity attribute of statistics in the 2010-census records is determined according to the administrative unit of the surveyed population. We also have made it clear in this section that if a residence is from a village, then the related statistics are aggregated into rural urbanity level; and if from a town, then it is township level; if from a city, it is urban level". Therefore, compared with "urban" and "township", the word "rural" only refers to those less developed/populated area within a province.

Moreover, I also see many pixels (with some built-up land) that are not assigned to any of the three grid types, but in Figure 4, buildings nearly spread all over the city of Shanghai, which confuses me a bit.

**Response:** Thanks for pointing this out. This difference is related to the setting of the geometry type of the visualization layer. In Figure 2 of the manuscript, in which only the Baoshan district of Shanghai is shown, the original point layer (Figure R1) has been transferred to polygon layer with grids of 0.009°×0.009°, approximate to 1km×1km resolution (Figure R2). One point in Figure R1 corresponds to one square in Figure R2. That is why some area is not assigned to any square.

While in Figure 4 of the manuscript, we show the whole Shanghai City, in which the visualization layer remains to be a point layer, only the symbology is set as square, thus it seems buildings spreads all over the whole city. But if Figure 4 is enlarged (as shown in Figure R3), there will also be gaps between grids.

Such visualization difference will not affect the calculation of exposure and seismic risk, since such spatial analyses usually need to calculate how many grids are located within specific area.

[Figure]

Figure R1: The original point layer of Figure 2 in the manuscript.

[Figure]

Figure R2: Converting the point layer of to polygon layer.

[Figure]

Figure R3: The enlarged version of point layer in Figure 4 of the manuscript , in which the symbology is set as square.

**Minor issues:**

1. Abstract part is too lengthy. The research background, method, result, and possibly implication need to be clearly stated, I suggest to remove some unnecessary details to enhance readability.

**Response:** Thanks for this suggestion. The abstract has been shortened by reducing the original **529** words to **344** words. The revised version of the abstract is as follows:

"To enhance the estimation accuracy of economic loss and fatality in seismic risk assessment, a high-resolution building exposure model is important. Previous studies in developing global and regional building exposure models usually use coarse administrative level (e.g., country, or sub-country level)

*census data as model inputs, which cannot fully reflect the spatial heterogeneity of buildings in large countries like China. To develop a high-resolution residential building stock model for mainland China, this paper uses finer urbanity level population and building-related statistics extracted from the records in Tabulation of the 2010 Population Census of the People's Republic of China (hereafter abbreviated as the "2010-census"). In the 2010-census records, for each province, the building-related statistics are categorized into three urbanity levels (urban, township, and rural). To disaggregate these statistics into high-resolution grid level, we need to determine the urbanity attributes of grids within each province. For this purpose, the geo-coded population density profile (with 1km×1km resolution) developed in the 2015 Global Human Settlement Layer (GSHL) project is selected. Then for each province, the grids are assigned with urban/township/rural attributes according to the population density in the 2015 GHSL profile. Next, the urbanity level building-related statistics can be disaggregated into grids, and the 2015 GHSL population in each grid is used as the disaggregation weight. Based on the four structure types (steel/reinforced-concrete, mixed, brick/wood, other) and five storey classes (1, 2-3, 4-6, 7-9, ≥10) of residential buildings classified in the 2010-census records, we reclassify the residential buildings into 17 building subtypes attached with both structure type and storey class and estimate their unit construction prices. Finally, we develop a geo-coded 1km×1km resolution residential building*

*exposure model for 31 provinces of mainland China. In each 1km×1km grid, the floor areas of the 17 residential building subtypes and their replacement values are estimated. The model performance is evaluated to be good and its practicability in seismic risk assessment is also checked. Limitations of this paper and future improvement directions are discussed. The whole modeling process of this paper is fully reproducible, and all the modeled results are publicly accessible."*

2. How do you define high resolution? Is 1 km of high resolution? Here your modelled results are of 1 km resolution. As far as I know, even the 30-m Landsat imaginaries are claimed to be moderate resolution (see for example: https://www.montana.edu/spowell/documents/pdffiles/powell_jars.pdf).  In addition, MODIS, which stands for the Moderate Resolution Imaging Spectroradiometer, is also moderate resolution, of course.

**Response:** Thanks for this comment. For building exposure model, the 1km*1km resolution is relatively high when compared with models at administrative level. It is true that the remote sensing datasets can be of much higher resolution, but for building exposure model development, additional attributes (e.g., the building structure type, story class, seismic design level, construction year etc.) needs to be attached with the remote sensing data to develop such a model. However, these attributes data are usually of lower

resolution. That is why their final product, the building exposure model, when is of 1km*1km resolution, can be considered as a high-resolution model.

3. Two key publications on mapping buildings particularly for China are missing here: https://www.sciencedirect.com/science/article/pii/S0034425720302297

https://www-sciencedirect-com.vu-l.idm.oclc.org/science/article/pii/S016920462100150X

**Response:** Thanks for recommending these two publications of Li et al. (2020) and Liu et al. (2021). Based on your recommendation, we find another two related studies, namely Ji et al. (2020) and Cao and Huang (2021). We will briefly introduce four papers in Section 4 when discussing the future improvement directions for exposure model development.

The introduction of these studies will be given as follows:

*"In addition, Li et al. (2020) developed the first continental-scale dataset on 3D building structure (including building footprint, height, and volume) at 1km×1km resolution for Europe, the US by using random forest models fed with remote sensing and Synthetic Aperture Radar imagery data. Liu et al. (2021) developed the urban floor area map for mainland China at 130m×130m resolution based on high spatial resolution nighttime light LUOJIA 1-01 images, a population map and a single building dataset encompassing 71 cities. Ji et al. (2020) generated the 10m×10m resolution rural settlements in the Yangtze River Delta of China by using the multi-source remote sensing*

*datasets with the Google Earth Engine Platform. Cao and Huang (2021) proposed a multi-spectral, multi-view, and multi-task deep network (called M3Net) for building height estimation. They estimated the building height at a spatial resolution of 2.5m×2.5m for 42 Chinese cities. Comparison with the results in Li et al. (2020) indicated that the M3Net method in Cao and Huang (2021) can better alleviate the saturation effect of high-rise building height estimation than the random forest method used in Li et al. (2020)."*

**References:**

Cao, Y. and Huang, X.: A deep learning method for building height estimation using high-resolution multi-view imagery over urban areas: A case study of 42 Chinese cities, Remote Sensing of Environment, 264(2021), 112590, doi: 10.1016/j.rse.2021.112590, 2021.

Ji, H., Li, X., Wei, X., Liu, W., Zhang, L., and Wang, L.: Mapping 10-m Resolution Rural Settlements Using Multi-Source Remote Sensing Datasets with the Google Earth Engine Platform, Remote Sensing, 12(17), 2832, doi:10.3390/rs12172832, 2020.

Li, M., Koks, E., Taubenböck, H., and van Vliet, J.: Continental-scale mapping and analysis of 3D building structure, Remote Sensing of Environment, 245(2021), 111859, doi:10.1016/j.rse.2020.111859, 2020.

Liu, M., Ma, J., Zhou, R., Li, C., Li, D., and Hu, Y.: High-resolution mapping of

mainland China's urban floor area, Landscape and Urban Planning, 214(2021), 104187, doi:10.1016/j.landurbplan.2021.104187, 2021.

---

## Author Response (AR1)

Dear Editor Sven Fuchs,

We deeply appreciate your efforts in managing the whole review process. Below is our response to comments from the two reviewers. The line numbers mentioned in the response refer to those in the revised manuscript with marked changes.

Best wishes,

Danhua Xin, James Daniell, Hing-Ho Tsang, and Friedemann Wenzel

**Comments from RC1 and our responses:**

Comment on "Residential building stock modeling for mainland China targeted for seismic risk assessment" by Xin et al.

**Comment 1:** The authors present an interesting approach to achieve a nation-wide model for the building stock to be used in seismic risk assessment. Based on various statistical data and information derived from secondary sources and remotely-sensed data, they present a method to derive at a geo-coded 1km×1km resolution residential building exposure model for 31 provinces of mainland China. Moreover, based on a sensitivity analysis for one case study, the authors present possible sources of uncertainty in the results, and show how these may be decreased during future research efforts.

**Response:** Thank you very much for your time and efforts in reviewing this manuscript.

**Comment 2:** Overall, the paper is timely, and well-structured. The overall point of criticism is that the sole use of statistical data to derive at a "real world" building stock model neglects region-specific and very local impacts on the distribution and quality of assets, which may lead to systematic over- and underestimation in certain areas of the country.

**Response:** It is true that regional specific and very local impacts are not considered in this modeling process. This is inevitable and mainly limited by the difficulty in collecting such detailed data for specific regions, which are usually proprietary and not publicly accessible.

**Comment 3:** Nevertheless, I strongly believe that the method is worth being published so that the international research community can further refine the method and decrease inherent uncertainties.

**Response:** Thanks for this affirmation. Our detailed responses to your comments are as follows.

Some minor comments:

**Comment 4:** Line 152/153: something is missing here. Should be re-formulated.

**Response:** Thanks for pointing this out. The two reasons that the township/street level population generated by using the multi-variate regression method in Fu et al. (2014a) tends to overpredict the population density in a sparsely populated area and underpredict the population density in a densely populated area are provided as follows:

"The reasons for such discrepancies are that: (1) The population density developed for each land use type by using the multi-variate regression method is the average population density, thus the over/under prediction of the actual population density in certain areas is inevitable; (2) When applying the multi-variate regression method, no additional supplementary data (e.g., road density, nighttime light) is employed to adjust the level of development in different regions, which is necessary because the level of development is much higher than the average in certain places such as the downtown area of metropolitan cities like Shenzhen and Guangzhou."

We have added the above explanation to **Line 151-160**, **Page 4-5** of the revised manuscript.

**Comment 5:** Line 193/194: could be better formulated.

**Response:** Thanks for pointing this out. The previous expression "*The census for the year 2020 is just initiated and normally it takes around two years to publish the final surveyed data. Therefore, the current latest census data are for the year 2010*" has been rephrased as "*Detailed statistics for the year 2020 are not publicly accessible yet. Therefore, census data for the year 2010 will be used to*

*elaborate the modeling process*", which can be checked from **Line 198-199**, **Page 6** of the revised manuscript.

**Comment 6:** Line 270/271: please elaborate a bit more why the spatial coverage is limited.

**Response:** Thanks for pointing this out. The limited spatial coverage of PopGrid China is due to an assumption in its development method, namely the multi-variate regression method (Fu et al., 2014a). It was assumed that the spatial distribution of population is limited to the six land use types recognized from the Landsat TM images, namely, cultivated land, forest land, grass land, rural residential land, urban residential land, and industrial and transportation land. However, in reality, the population is distributed more widely beyond these six land use types.

We have added the above explanation to **Line 278-282**, **Page 8** of the revised manuscript.

**Comment 7:** Line 469: in times of almost unlimited computing capacity, this should not be an issue. In contrast, applying the same unit price over the entire area of (mainland) China is a major source of uncertainty of the method, which should be addressed in more detail in the respective section 4.

**Response:** Thanks for pointing this out. In Line 467-469, we emphasized that "*There are significant differences across the country in terms of economic development level, geographic and climatic diversity, and standardization in building construction. Therefore, it is mainly for computational convenience that this paper applies the same unit construction price for all the provinces and all the urbanity levels.*". The "computational convenience" here does not refer to the limitation in computing capacity of hardware. Instead, we mean it is convenient to only use a uniform price list to preliminarily calculate the replacement value of residential buildings in each grid, since it is difficult to compile a complete and accurate building construction price list for different provinces and urbanities. On one hand, this is because different documents include different items when calculating their unit construction prices. For example, some consider the cost to build the main structure only, while others consider also the cost of supporting facility and landscaping. On the other hand, different areas have different seismic

design requirement, which will also alter the unit construction price for the same type of building.

Actually, before compiling the unit construction price list in Table 4 of the manuscript, we have consulted cost-engineers from the real-estate industry. Their internal documents for cost control indicate that for the same type of residential building, the difference in unit construction price of the main structure in different regions in China is limited to 300 RMB. Also, according to Li et al. (2021), the average unit construction price of multi-story reinforced concrete buildings in urban areas of Tibet reaches 3200 RMB/m$^2$, which is comparable to the unit construction price of the same type of building in those more developed coastal areas.

Therefore, we prefer to only provide a reference set of unit construction price in Table 4 and avoid to over-manipulate it. As you point out, this will cause major uncertainty and we totally agree. However, this simplicity and transparency will make it more convenient for potential model users to adjust the reference price to their targeted study area by simply applying some rectification factors.

**Comment 8:** Section 3.1.2: Here it is not clear to me what the key message is; obviously, higher buildings will have a higher density of floor area and, thus, a higher population density. My suggestion is to elaborate this a bit more, or to delete this particular section.

**Response:** Thanks for this comment. As implied by the title of this subsection, this short section serves as an example to demonstrate the modeled floor area in Shanghai, which can help potential readers to conduct direct comparison with other reports or modeling results. For example, readers can directly check whether the grids that have the highest modeled floor area are within those most prosperous regions in Shanghai. It is our assumption that grids with higher population are more "urban". We agree with your comment that obviously higher buildings will have a higher density of floor area and, thus a higher population density. Therefore, we delete panel (c) in the revised version of Figure 4.

According to the above explanation, we slightly re-elaborated this section – refer **Line 483-489**, **Page 13** of the revised manuscript.

**Comment 9:** Figure 1: technically, the classes are not clearly distinguishable (what if a grid has exactly 4936 or 2750 inhabitants?), please adjust.

**Response:** Thanks for pointing this out. We think you refer to Figure 2. The modified population ranges in the legend for urban/township/rural urbanity levels are given in the revised version of Figure 2, in which the population number of 4936 and 2750 is attributed to urban and township level, respectively. In addition, the size of grid is changed from 0.009°×0.009° to 0.011°×0.0087° to eliminate the gaps generated during the data processing process in ArcGIS. The modified grid size is also approximate to 1km×1km resolution.

[Figure]

Figure 2: Revised version. The grid size changes from 0.009°×0.009° to 0.011°×0.0087°, which is also approximate to 1km×1km resolution.

**Comment 10:** Figure 4 (and related section in the main text body): from my understanding it would be more explanatory how well your method is suitable for application if you would compute the differences between the modelled floor area per km^2 and the 3d view provided in inlet (c), also in terms of uncertainty quantification. Please also consider the similar issue of classes given already for Figure 1 (and check all the other Figures, also in Figure 9 this is wrong).

**Response:** We are afraid there is misunderstanding here. Indeed, the population data in Figure 2 (not Figure 1) and old Figure 4(c) are from the 2015 GHSL developed by the JRC (Joint Research Center, European Commission). And this population density profile is the base for us to divide the grids in each province into urban/township/rural levels. After this, we further disaggregate 2010-census statistics for urban/township/rural levels into corresponding grids.

Therefore, Figure 2 is to demonstrate how we assign urban/township/rural attributes to grids according to 2015 GHSL population density. Figure 4(a) is the example demonstrating one of our modeled products, namely the floor area in each grid. In old Figure 4(c), its height is determined by the 2015 GHSL population in these grids, but the floor area is the same as that in Figure 4(a). By plotting old Figure 4(c), we mainly wanted to show the good correlation between floor area and population, which you think is obvious and we agree. Therefore, in the revised manuscript, we remove panel (c) of Figure 4.

Figure 9 is quite different from Figure 2 or Figure 4, because it is an application of the modeled results. It gives the seismic loss ratios calculated based on our modeled building floor area, replacement value as well as an empirical vulnerability curve and the intensity map of the 2008 Ms8.0 Wenchuan earthquake.

**Comment 11:** Given these constraints I recommend revisions before the manuscript may become acceptable for publication.

**Response:** We deeply appreciate your generosity in spending time on reviewing this manuscript, which requires a lot of patience and efforts. We hope our responses have adequately addressed your concerns.

**References mentioned in the responses to RC1:**

Fu, J., Jiang, D. and Huang, Y.: Populationgrid_China, Acta Geographica Sinica, 69(Supplement), 41–44, doi: 10.11821/dlxb2014S006, 2014a (in Chinese).

Li, C., Li, Z., Lyu, H., and Gao, M.: Probabilistic seismic risk assessment for the Eastern Himalayas, China, Earthquake Spectra, doi: 10.1177/8755293021999056, 2021.

**Comments from RC2 and our responses:**

This paper basically proposed a downscaling approach to allocating building stock per province in China for better risk assessment, which well fits the scope of this journal. Given current revision status of this paper, I only have one main concern, which is about the definition of urban/township/rural. In reality, we can hardly differentiate them, in particular at pixelated level. Shanghai as we all know is highly urbanized, but still I can see many rural pixels from Figure 1, I do not believe it is the real case here. To me, 'rural' is mostly remote natural areas, I assume you intend to say 'village'.

**Response:** Thanks for your review. We totally understand your concern. It is true that in those highly urbanized provinces (e.g., Shanghai, Beijing, Guangzhou, Zhejiang, Jiangsu etc.), it is difficult to imagine the existence of rural areas. However, in Shanghai, except those most developed districts (i.e. those included in the downtown area of Figure 4 and part of the Pudong, Minhang and Baoshan districts), many places of the remaining districts, including Jiading, Qingpu, Songjiang, Jinshan, Fengxian, indeed belong to countryside or rural area. Also, from the building related statistics extracted from the 2010-census records for Shanghai (Table R1), we can clearly see that the numbers of families living in 1, 2-3 storey classes at rural level are comparable to those at township and urban levels. The numbers of families living in "mixed masonry", "brick/wood" and "other" structure types at rural level are also comparable to those at township and/or urban levels.

Table R1: The building related statistics of Shanghai from 2010-census records.

| urbanity | Number of families grouped by storey class | | | | | Number of families grouped by structure type | | | |
|---|---|---|---|---|---|---|---|---|---|
| | 1 | 2-3 | 4-6 | 7-9 | ≥10 | steel/RC | mixed masonry | brick/wood | other |
| urban | 60506 | 116799 | 304794 | 27780 | 104766 | 268377 | 249438 | 93734 | 3096 |
| township | 24233 | 44272 | 29262 | 638 | 4710 | 35992 | 46750 | 19423 | 950 |
| rural | 31644 | 57352 | 3415 | 49 | 264 | 8884 | 48551 | 33963 | 1326 |

In addition, this webpage (https://www.zhihu.com/question/301964832) collects the pictures of countryside area in Shanghai shared by people living there, which is not so different from rural area of other provinces (the webpage can be directly translated into English by Google Translator).

We consider that it may not be so appropriate to change "rural" to "village" in the context of this article, since villages typically have administrative boundaries. As explained in Section 2.3 of the manuscript, "*The urbanity attribute of statistics in the 2010-census records is determined according to the administrative unit of the surveyed population.*" We have also explained in this section that the division of urban/township/rural level in the 2010-census is based on the administrative belongings of the surveyed population. If a residence is from a village, then the related statistics are aggregated into rural urbanity level; and if from a town, then it is township level; if from a city, it is urban level. Therefore, compared with "urban" and "township", the word "rural" only refers to those less developed/populated area within a province.

Moreover, I also see many pixels (with some built-up land) that are not assigned to any of the three grid types, but in Figure 4, buildings nearly spread all over the city of Shanghai, which confuses me a bit.

**Response:** Thanks for pointing this out. This difference is related to the setting of the geometry type of the visualization layer in ArcGIS. In Figure 2 of the manuscript, in which only the Baoshan district of Shanghai is shown, the original point layer (Figure R1) has been transferred to polygon layer with grid size of 0.009°×0.009°, approximate to 1km×1km (Figure R2). One point in Figure R1 corresponds to one square in Figure R2. That is why some area is not assigned to any square.

Meanwhile, Figure 4 of the manuscript shows the whole Shanghai City, in which the visualization layer remains to be a point layer, whereas the symbol is set as square. Thus, it seems that buildings spread across the whole city. But if Figure 4 is enlarged (as shown in Figure R3), there will be gaps between grids.

[Figure]

Figure R1: The original point layer of Figure 2 in the manuscript.

[Figure]

Figure R2: Converting the point layer to polygon layer.

[Figure]

Figure R3: The enlarged version of point layer in Figure 4 of the manuscript, in which the symbol is set as square.

To avoid further misunderstanding due to gaps among grids caused by the transformation of point layer to polygon layer in ArcGIS, we replot Figure 2 and Figure 4 (see below), in which the size of the grid changes from 0.009°×0.009° to 0.011°×0.0087°, which is also approximate to 1km×1km.

[Figure]

Figure 2: Revised version. The grid size changes from the old size of 0.009°×0.009° to 0.011°× 0.0087°, which is also approximate to 1km×1km resolution.

[Figure]

Figure 4: Revised version. The grid size in Figure 4(a) changes from the old size of 0.009°×0.009° to 0.011°×0.0087°, which is also approximate to 1km×1km. The old panel (c) is deleted. Note that the legend in Figure 4 is different from that in Figure 2.

**Minor issues:**

1. Abstract part is too lengthy. The research background, method, result, and possibly implication need to be clearly stated, I suggest removing some unnecessary details to enhance readability.

**Response:** Thanks for this suggestion. The abstract has been shortened from **529** words to **347** words. The revised version of the abstract is as follows:

"To enhance the estimation accuracy of economic loss and casualty in seismic risk assessment, a high-resolution building exposure model is necessary. Previous studies in developing global and regional building exposure models usually use coarse administrative level (e.g., country, or sub-country level) census data as model inputs, which cannot fully reflect the spatial heterogeneity of buildings in large countries like China. To develop a high-resolution residential building stock model for mainland China, this paper uses finer urbanity level population and

*building-related statistics extracted from the records in Tabulation of the 2010 Population Census of the People's Republic of China (hereafter abbreviated as the "2010-census"). In the 2010-census records, for each province, the building-related statistics are categorized into three urbanity levels (urban, township, and rural). To disaggregate these statistics into high-resolution grid level, we need to determine the urbanity attributes of grids within each province. For this purpose, the geo-coded population density profile (with 1km×1km resolution) developed in the 2015 Global Human Settlement Layer (GSHL) project is selected. Then for each province, the grids are assigned with urban/township/rural attributes according to the population density in the 2015 GHSL profile. Next, the urbanity level building-related statistics can be disaggregated into grids, and the 2015 GHSL population in each grid is used as the disaggregation weight. Based on the four structure types (steel/reinforced-concrete, mixed, brick/wood, other) and five storey classes (1, 2-3, 4-6, 7-9, ≥10) of residential buildings classified in the 2010-census records, we reclassify the residential buildings into 17 building subtypes attached with both structure type and storey class and estimate their unit construction prices. Finally, we develop a geo-coded 1km×1km resolution residential building exposure model for 31 provinces of mainland China. In each 1km×1km grid, the floor areas of the 17 residential building subtypes and their replacement values are estimated. The model performance is evaluated to be satisfactory and its practicability in seismic risk assessment is also confirmed. Limitations of the proposed model and directions for future improvement are discussed. The whole modeling process presented in this paper is fully reproducible, and all the modeled results are publicly accessible."*

2. How do you define high resolution? Is 1 km of high resolution? Here your modelled results are of 1 km resolution. As far as I know, even the 30-m Landsat imaginaries are claimed to be moderate resolution (see for example: https://www.montana.edu/spowell/documents/pdffiles/powell_jars.pdf). In addition, MODIS, which stands for the Moderate Resolution Imaging Spectroradiometer, is also moderate resolution, of course.

**Response:** Thanks for this comment. For building exposure model targeted for seismic risk assessment, 1km×1km resolution is relatively high when compared

with models at administrative level (e.g., Bal et al., 2010; Dabbeek et al., 2021). It is true that remote sensing datasets can be of much higher resolution, but for building exposure model development, additional attributes (e.g., the building structure type, story class, seismic design level, construction year, etc.) need to be attached with the remote sensing data. However, data for these attributes are usually of much lower resolution, especially for large research area (like here for mainland China). That is why their final product, the building exposure model, which is of 1km×1km resolution, can be considered as a high-resolution model.

3. Two key publications on mapping buildings particularly for China are missing he re: https://www.sciencedirect.com/science/article/pii/S0034425720302297, https://www-sciencedirect-com.vu-l.idm.oclc.org/science/article/pii/S016920462100150X

**Response:** Thanks for recommending the two publications of Li et al. (2020) and Liu et al. (2021). Based on your recommendation, we find two other related studies, namely Ji et al. (2020) and Cao and Huang (2021). We have briefly reviewed these four papers in Section 4 of the revised manuscript (**Line 621-632**, **Page 17**) when discussing the future improvement directions for exposure model development. The added paragraph is copied below:

*"In addition, Li et al. (2020) developed the first continental-scale dataset on 3D building structure (including building footprint, height, and volume) at 1km×1km resolution for Europe, China, and the US by using random forest models fed with remote sensing and Synthetic Aperture Radar imagery data. Liu et al. (2021) developed the urban floor area map for mainland China at 130m×130m resolution based on high spatial resolution nighttime light LUOJIA 1-01 images, a population map and a single building dataset encompassing 71 cities. Ji et al. (2020) generated the 10m×10m resolution model of rural settlements in the Yangtze River Delta of China by using the multi-source remote sensing datasets with the Google Earth Engine Platform. Cao and Huang (2021) proposed a multi-spectral, multi-view, and multi-task deep network (called M3Net) for building height estimation. They estimated the building height at a spatial resolution of 2.5m×2.5m for 42 Chinese cities. Comparison with the results in Li et al. (2020) indicated that the M3Net method in Cao and Huang (2021) can better alleviate*

*the saturation effect of high-rise building height estimation than the random forest method used in Li et al. (2020)."*

**References:**

Bal, I. E., Bommer, J. J., Stafford, P. J., Crowley, H., and Pinho, R.: The Influence of Geographical Resolution of Urban Exposure Data in an Earthquake Loss Model for Istanbul, Earthquake Spectra, 26, 619–634, doi:10.1193/1.3459127, 2010.

Cao, Y. and Huang, X.: A deep learning method for building height estimation using high-resolution multi-view imagery over urban areas: A case study of 42 Chinese cities, Remote Sensing of Environment, 264(2021), 112590, doi: 10.1016/j.rse.2021.112590, 2021.

Dabbeek, J., Crowley, H., Silva, V., Weatherill, G., Paul, N., and Nievas, C. I.: Impact of exposure spatial resolution on seismic loss estimates in regional portfolios, Bulletin of Earthquake Engineering, doi:10.1007/s10518-021-01194-x, 2021.

Ji, H., Li, X., Wei, X., Liu, W., Zhang, L., and Wang, L.: Mapping 10-m Resolution Rural Settlements Using Multi-Source Remote Sensing Datasets with the Google Earth Engine Platform, Remote Sensing, 12(17), 2832, doi:10.3390/rs12172832, 2020.

Li, M., Koks, E., Taubenböck, H., and van Vliet, J.: Continental-scale mapping and analysis of 3D building structure, Remote Sensing of Environment, 245(2021), 111859, doi:10.1016/j.rse.2020.111859, 2020.

Liu, M., Ma, J., Zhou, R., Li, C., Li, D., and Hu, Y.: High-resolution mapping of mainland China's urban floor area, Landscape and Urban Planning, 214(2021), 104187, doi:10.1016/j.landurbplan.2021.104187, 2021.